# A complex regulatory landscape involved in the development of mammalian external genitals

Ana Rita Amândio[1], Lucille Lopez-Delisle[1], Christopher Chase Bolt[1], Bénédicte Mascrez[2], Denis Duboule[1,2,3]*

[1]School of Life Sciences, Ecole Polytechnique Fédérale de Lausanne (EPFL), Lausanne, Switzerland; [2]Department of Genetics and Evolution, University of Geneva, Geneva, Switzerland; [3]Collège de France, Paris, France

**Abstract** Developmental genes are often controlled by large regulatory landscapes matching topologically associating domains (TADs). In various contexts, the associated chromatin backbone is modified by specific enhancer–enhancer and enhancer–promoter interactions. We used a TAD flanking the mouse *HoxD* cluster to study how these regulatory architectures are formed and deconstructed once their function achieved. We describe this TAD as a functional unit, with several regulatory sequences acting together to elicit a transcriptional response. With one exception, deletion of these sequences didn't modify the transcriptional outcome, a result at odds with a conventional view of enhancer function. The deletion and inversion of a CTCF site located near these regulatory sequences did not affect transcription of the target gene. Slight modifications were nevertheless observed, in agreement with the loop extrusion model. We discuss these unexpected results considering both conventional and alternative explanations relying on the accumulation of poorly specific factors within the TAD backbone.

*For correspondence:
denis.duboule@epfl.ch

Competing interests: The authors declare that no competing interests exist.

## Introduction

During mammalian development, the organization of body structures and their morphogenesis require the accurate transcriptional regulation of the *Hox* gene family of transcription factors. These proteins instruct progenitor cells at different levels along the main anterior to posterior axis, about their developmental fates. In addition to this ancient role in trunk patterning, subsets of the four *Hox* gene clusters were co-opted during evolution to promote the development of secondary body axes such as the limbs and the external genitalia (*Dollé et al., 1991a*). In the latter case, mice lacking both *Hoxa13* and *Hoxd13* functions fail to develop external genitalia due to a complete agenesis of the genital tubercle (GT) (*Kondo et al., 1997*; *Warot et al., 1997*).

In the case of the *HoxD* cluster, the control of gene transcription in the emerging GT involves *cis*-regulatory sequences located in a 700 kb large regulatory landscape positioned 5' to the cluster, referred to as C-DOM (centromeric domain)(*Andrey et al., 2013*; *Montavon et al., 2011*; *Spitz et al., 2003*). This regulatory landscape matches one of the two topologically-associating domains (TADs), which flank the gene cluster. The functional importance of C-DOM was confirmed in vivo using chromosome engineering. For example, when this region was repositioned 3 Mb away from *HoxD*, transcription of *Hoxd13* in the GT was almost entirely abolished (*Tschopp and Duboule, 2011*) and subsequent deletions spanning various parts of C-DOM supported this conclusion (*Lonfat et al., 2014*). Genetic and biochemical analyses have shown that this entire regulatory landscape is shared between GT and digits, and contains multiple enhancer sequences that are active in either both or only one of these developing structures (*Gonzalez et al., 2007*; *Lonfat et al., 2014*; *Montavon et al., 2011*). Overall, it appears that within a large constitutive TAD structure, subtle yet

specific modifications of chromatin architecture are formed either in GT or in digit cells (*Lonfat and Duboule, 2015*).

Unlike the regulatory landscape located at the opposite side of the *HoxD* cluster (T-DOM), which includes a large variety of enhancers with distinct specificities regulating 'anterior' *Hoxd* genes, the C-DOM appears to be devoted to the control of the most posterior and distal terminal body structures by regulating mostly *Hoxd13* either in digit cells or in the GT. The tropism of C-DOM enhancers for *Hoxd13* results from the presence of a strong chromatin boundary between this target gene and the rest of the cluster, which concentrates the action of centromeric enhancer on this precise target (*Rodríguez-Carballo et al., 2017*). Over the past years, the importance of the C-DOM in controlling *Hoxd* genes expression has been clearly demonstrated. However, both the dynamic behavior of such a regulatory landscape that is its implementation and decommissioning, as well as the functional contribution of specific *cis*-elements in these processes remained to be established. This is necessary to understand how an entire TAD can be transcriptionally mobilized in different morphogenetic contexts to achieve similar regulatory outcomes. A 'specific' view of the regulatory system would involve discriminative factors, progressively building a tissue-specific chromatin context with a deterministic strategy. Alternatively, a more generic process could be considered, where the accumulation of various factors available in different tissues would elicit the same transcriptional response through whichever chromatin configuration they would trigger.

In this work, we studied the chromatin conformation dynamics at the *HoxD* locus during GT development, as well as the functional contribution of specific *cis*-elements to *Hoxd* genes regulation. We observed that the gross chromatin organization of C-DOM predates the appearance of the GT. As GT development progresses, we scored a reduction in transcript levels correlating with a decrease in enhancer-promoter chromatin loops within C-DOM. This decrease occurred while maintaining a subset of CTCF associated contacts, which are preserved independently from the transcriptional status of the gene cluster. While both the deletion of the *Prox* enhancer and deletions of clusters of enhancers severely affected *Hoxd* genes transcript levels, the deletions of most enhancers in isolation had little (if any) effect on transcription in the GT. Moreover, the deletion of the only bound CTCF site detected in the central part of the regulatory landscape, did not impact the transcriptional outcome, even though its inversion reallocated contacts in a manner compatible with the loop extrusion model (*Fudenberg et al., 2016*; *Rao et al., 2014*; *Vian et al., 2018*). These results point to a high resilience of the regulatory strategy at work in this locus. They also suggest the existence in the same TAD of distinct mechanisms to control target gene activation, either relying upon sequence specific enhancer-promoter interactions, or involving less deterministic parameters and using the underlying chromatin structure.

## Results

### *Hox* genes and GT development

To assess *Hox* genes transcription during GT development, we initially quantified their expression levels by using RNA-sequencing (RNA-seq) and analyzed datasets from three different stages of GT embryonic development starting from embryonic day 12.5 (E12.5), E16.5 and E18.5. We observed that genes positioned in the 5′ portion of both *HoxA* (*Hoxa7* to *Hoxa13*) and *HoxD* (*Hoxd8* to *Hoxd13*) clusters were expressed at all developmental stages (*Figure 1A*, *Figure 1—figure supplement 1* and *Figure 1—source data 1*). Furthermore, with the exception of *Hoxc11* and *Hoxc10*, only basal levels of mRNAs were scored for the *HoxC* and *HoxB* clusters (*Figure 1—figure supplement 1* and *Figure 1—source data 1*), consistent with previous observations (*Hostikka and Capecchi, 1998*; *Montavon et al., 2008*). Overall, we detected a general decrease in the amount of *Hox* mRNAs during GT development, in particular for *Hoxd12* and *Hoxd13* (*Figure 1A*).

To describe the dynamics of *Hoxd* transcript accumulation during GT development, we micro-dissected the cloaca region (CR) at E10.5, the major contributing embryonic tissue to the emergence of the GT (*Georgas et al., 2015*), as well as genital buds at E12.3, E13.5, E15.5, E16.5, E17.5 and E18.5. We performed RT-qPCR for *Hoxd13* and detected transcripts in the CR at E10.5 (*Figure 1B* and *Figure 1—source data 1*), followed by a significant increase in transcript levels between the CR and the E12.5 GT (p<0.0001). The mRNA levels then remained constant between E12.5 and E13.5, whereas they were significantly reduced in the interval between E13.5 and E15.5 GTs (p<0.0001).

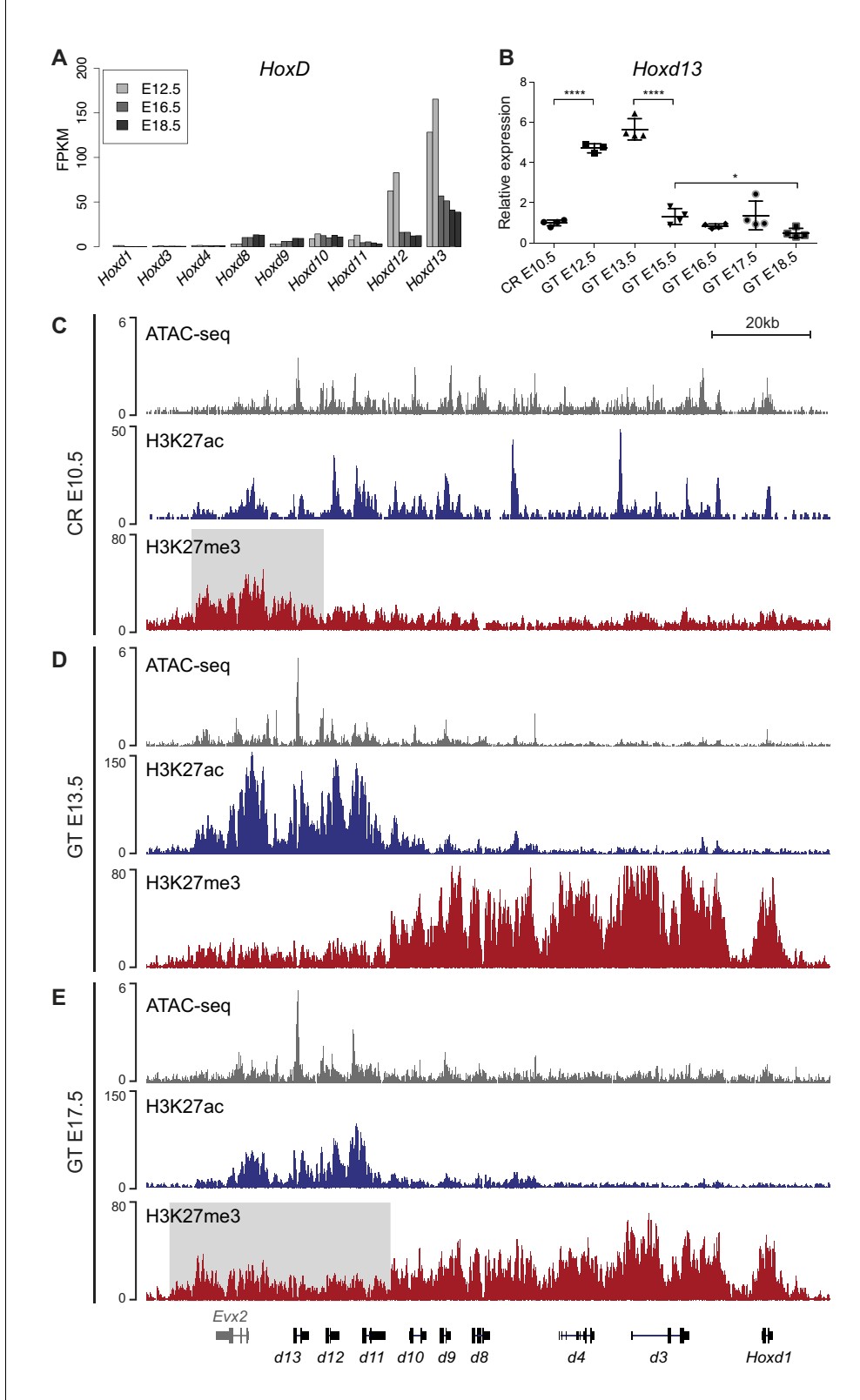

**Figure 1.** Transcription of *Hoxd* genes in developing GT. (**A**) Quantification of *Hoxd* genes transcript levels by RNA-seq (FPKM values) in GT at E12.5 (*Amândio et al., 2016*), E16.5 and E18.5. (**B**) RT-qPCR of *Hoxd13* mRNAs in different stages of GT development. The plotted values indicate the ratio of expression using the cloaca region (CR) as a reference (n ≥ 3 biological replicates for each sample). A Welch's *t*-test was used to evaluate the putative

*Figure 1 continued on next page*

*Figure 1 continued*

significant changes in *Hoxd13* expression. Bars indicate mean with SD, ****p<0.0001, *p=0.0175. (C–E) ATAC-seq (gray) and ChIP-seq profiles for H3K27ac (blue) and H3K27me3 (red) at the *HoxD* locus in E10.5 wildtype CR (C), E13.5 GT (D) and E17.5 GT (E). Coordinates (mm10): chr2:74637433–74775728. The gray box in track three indicates the enrichment of H3K27me3 at 5'-located *Hoxd* genes in the CR. The gray box in track nine indicates the relative gain of H3K27me3 at 5'-located *Hoxd* genes in E17.5 GT when compared to the E13.5 GT sample. The online version of this article includes the following source data and figure supplement(s) for figure 1:

**Source data 1.** *Hox* genes expression values (FPKM and RT-qPCR) during GT development.
**Figure supplement 1.** *Hox* genes expression profile during GT development.

After E15.5, the transcript levels continued to decrease yet to a lesser extent (between E15.5 and E18.5; p=0.0175, *Figure 1B* and *Figure 1—source data 1*), confirming the RNA-seq results (*Figure 1A*).

We next compared chromatin accessibility and selected histone modifications in three developmental stages to correlate with transcript levels. We used the CR at E10.5 (prior to GT formation; low *Hoxd13* expression), GT at E13.5 (early GT development; high *Hoxd13* expression) and GT at E17.5 (late GT development; low *Hoxd13* expression) and performed ATAC-seq and ChIP-seq for both H3K27ac and H3K27me3 chromatin marks. At E10.5, prior to GT formation, all *Hoxd* genes and *Evx2* were accessible as defined by ATAC-seq (*Figure 1C*). H3K27ac signals of moderate intensity were scored over the *Hoxd9* to *Evx2* DNA interval as well as peaks over the promoters of *Hoxd1*, *Hoxd3,* and *Hoxd4* (*Figure 1C*), indicating a somewhat general activity of *Hoxd* genes in this posterior region of the elongating body axis. This was confirmed by a low coverage in H3K27me3 marks, which were detected mostly over the *Evx2* gene, next to the *Hox* cluster (*Figure 1C*, gray area).

In the growing E13.5 genital bud, a different picture was observed with a whole inactivation of the cluster from *Hoxd1* to *Hoxd10-11*, as indicated by a robust coverage of this region by H3K27me3 marks and the disappearance of both H3K27ac marks and ATAC-seq signals (*Figure 1D*). In contrast, ATAC-seq peaks remained in the *Hoxd11* to *Evx2* region, accompanied by a large increase in H3K27ac signals (*Figure 1D*) reflecting full transcription of the latter genes. At this stage, a clear separation of the cluster into two distinct epigenetic domains was scored, reminiscent of the situation described in distal forelimb buds (*Andrey et al., 2013*). At E17.5, this clear dichotomy between epigenetic domains, as labelled by the H3K27ac and ATAC-seq signals, was still detected though at a lower magnitude (*Figure 1E*). In parallel, H3K27me3 marks started to spread over the entire gene cluster (*Figure 1E*). These data are in agreement with the analysis of mRNA levels as observed by both RNA-seq and RT-qPCR.

## Implementation and decommissioning of a chromatin architecture

*Hoxd* genes are regulated in the developing GT by long-range acting sequences positioned within the flanking centromericly located TAD (C-DOM; *Figure 2A*). To assess the dynamics of the TAD structure during bud development, we used circularized chromosome conformation capture combined with high-throughput sequencing (4C-seq) to reveal the physical chromatin interactions established between *Hoxd13* and the C-DOM, at various developmental stages. *Hoxd13* was selected as a viewpoint since it is the highest expressed *Hoxd* gene in this tissue and because it is necessary for proper genital development. Indeed, while its disruption alone leads to alterations in external genitals, its combined inactivation with *Hoxa13* leads to a complete agenesis of genitals (*Dollé et al., 1993*; *Kondo et al., 1997*; *Warot et al., 1997*). We micro-dissected CR at E10.5 and GTs at E12.5, E13.5, E15.5, and E17.5, and used forebrain at E12.5 as a control tissue lacking *Hoxd* mRNA. As a baseline to our temporal series, we used a mouse embryonic stem cells (mESC) dataset (*Noordermeer et al., 2014*) assuming that these cells reflect the ground-state 3D architecture of the gene cluster.

In mESC, contacts between *Hoxd13* and the C-DOM were mainly scored in the island II and V regions (*Figure 2B*, top, red lines). A large proportion of the interactions was scored in the cluster itself where they were likely driven by H3K27me3 marks (*Vieux-Rochas et al., 2015*). This 3D architecture was altogether quite comparable to that found in forebrain cells, with discrete contacts

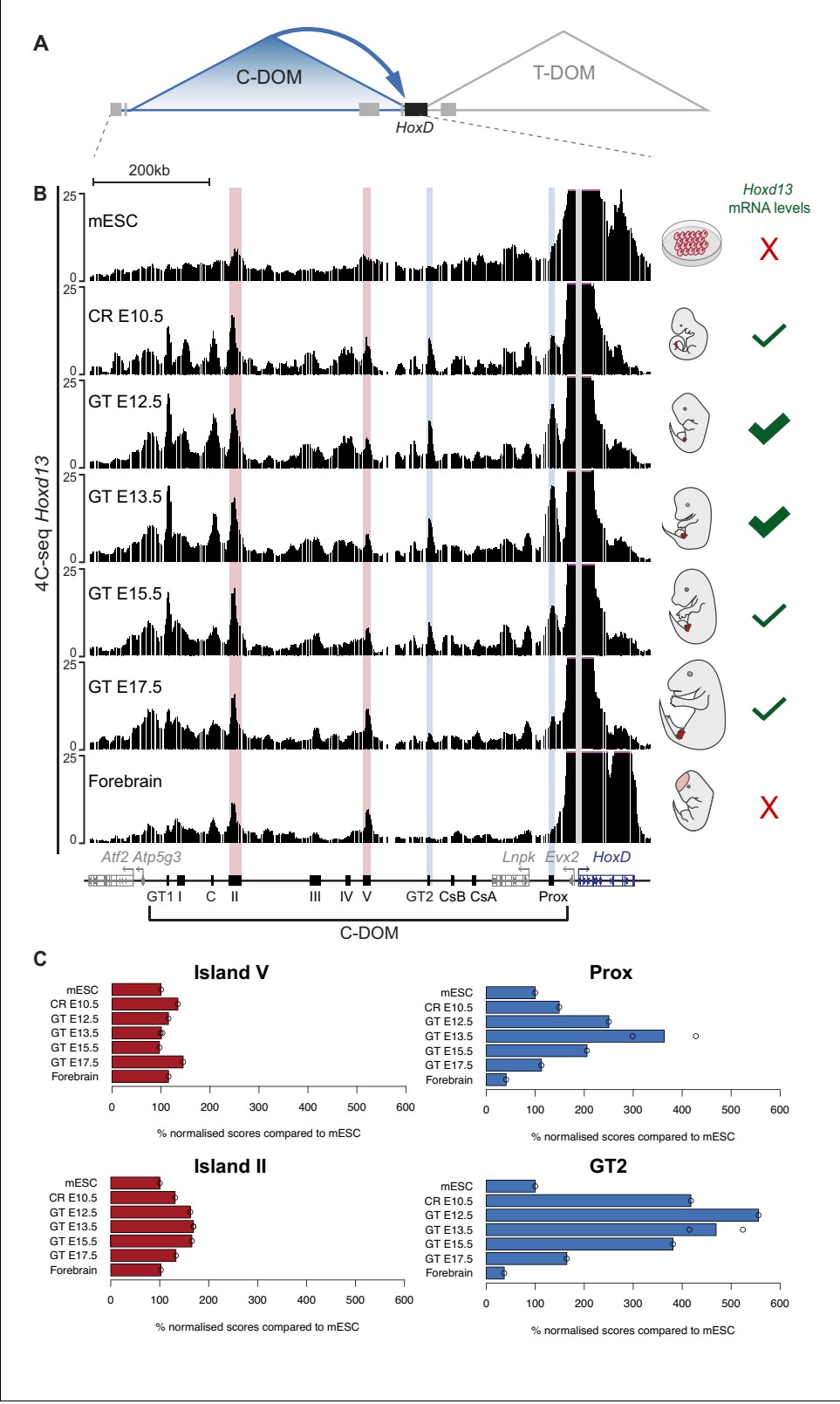

**Figure 2.** Chromatin topology of C-DOM during GT development. (**A**) Schematic representation of the two regulatory landscapes, with the centromeric (C-DOM) and telomeric (T-DOM) TADs flanking the *HoxD* cluster (black box), which acts as a TAD boundary. Gray boxes represent non-*Hox* genes. The *cis*-regulatory elements involved in the control of *Hoxd* gene transcription in the GT are located in C-DOM (blue arrow). (**B**) 4C-seq

*Figure 2 continued on next page*

*Figure 2 continued*

interactions profiles between the *Hoxd13* viewpoint (gray line) and both the *HoxD* cluster and the C-DOM. From top to bottom, 4C-seq profiles from mouse ES cells (mESC; track 1) (*Noordermeer et al., 2014*), E10.5 CR, E12.5 GT, E13.5 GT, E15.5 GT, E17.5 GT and fetal forebrain cells (track 7) are represented. Coordinates (mm10): chr2:73815520–74792376. A schematic representation of the *HoxD* cluster and the C-DOM is shown below with known enhancers as black boxes. The vertical blue lines highlight changes in chromatin interactions between *Hoxd13* and both Prox and GT2 in the different developmental stages and tissues analyzed. Vertical red lines highlight the contacts between *Hoxd13* and island II or island V, which remained fairly constant in all samples analyzed. The mRNA levels of *Hoxd13* in each condition are schematized on the right side of the corresponding profile. (C) Bar plots showing the quantification of the ratio of the number of normalized reads (+/- 5 Mb around the viewpoint) in selected regulatory regions, using mouse ES cells as a reference. Open circles represent each individual replicate. The regulatory element analyzed is indicated on top of each plot.

The online version of this article includes the following figure supplement(s) for figure 2:

**Figure supplement 1.** Chromatin interactions in C-DOM during GT development.

established between *Hoxd13* and island II and V (*Figure 2B*, bottom, red lines) and hence we estimate that these profiles reflected the 3D chromatin state of C-DOM in the complete absence of transcription. Upon transcriptional activation, however, frequencies of contacts with the C-DOM increased and interactions between *Hoxd13* and previously characterized enhancers (Prox, GT2) (*Gonzalez et al., 2007*; *Lonfat et al., 2014*) became visible (*Figure 2B*, second track, blue lines). This dataset showed a C-DOM specific chromatin architecture that is organized before the emergence of the genital bud.

In subsequent stages of GT development (E12.5 or E13.5), contacts between various enhancer regions and *Hoxd13* continued to increase when compared to the CR sample (*Figure 2B*). As development further progressed, contacts established between *Hoxd13* and C-DOM weakened. At the latter stage the profile observed was comparable to either forebrain cells or the mESC profiles, with a loss of contacts with specific enhancers (Prox and GT2; *Figure 2B*, blue lines). We quantified the percent of fragments covering each regulatory island by using mESC as a reference (*Figure 2C*). The relative frequency of contacts with island II and island V remained fairly constant in all samples analyzed. In contrast, the contacts between *Hoxd13* and either Prox or GT2 dramatically increased from the mESC to the E13.5 GT samples. The decrease in contacts observed between E13.5 to E17.5 GTs correlated with a decrease in *Hoxd13* transcript levels. Fetal forebrain cells, which do not express any *Hoxd* genes, showed the lowest values of interactions between *Hoxd13* and either Prox or GT2 (*Figure 2C*).

To validate these results, we selected both the GT2 region, which displayed important changes in interaction frequencies with *Hoxd13* during GT development, and the island V region which showed more constitutive contacts, as viewpoints in 4C-seq experiments. We used 4C-seq libraries for E12.5, E13.5, E15.5, and E17.5 GT cells and E12.5 forebrain cells as the negative control. We confirmed that the interactions between GT2 and the *Hoxd13* region substantially decreased from E13.5 to E17.5, whereas contact frequencies between island V and *Hoxd13* was essentially stable, regardless of the stage and tissue analyzed (*Figure 2—figure supplement 1*). Therefore, as transcription decreased, some contacts established with C-DOM were lost whereas others were maintained (island II and island V), indicating that at the time transcription is switched off, C-DOM goes back to the pre-organized chromatin backbone that characterizes tissues or cells that do not express any *Hox* genes. Of note, the constitutively contacted island V includes a binding site occupied by CTCF (see below), a protein described to facilitate enhancer-promoter contacts by DNA-looping (see *Ong and Corces, 2014*), even though it does not always appear necessary in this process (*Bonev et al., 2017*; *Deng et al., 2012*; *Stadhouders et al., 2018*).

## Dissecting the regulatory potential of the C-DOM TAD

We next explored the functional dynamics of C-DOM during GT development. A detailed analysis of our CR ATAC-seq dataset revealed several accessible chromatin sites, some of which match previously identified GT enhancers such as GT2 (*Lonfat et al., 2014*; *Figure 3A*, black arrowhead). Noteworthy, the GT and limb enhancer sequence Prox was not yet accessible at this stage (*Figure 3A*, red arrowhead). At E13.5, when the C-DOM is fully active, several ATAC-seq peaks were scored over previously characterized enhancers within this region, including Prox and GT2 (*Figure 3A*). In

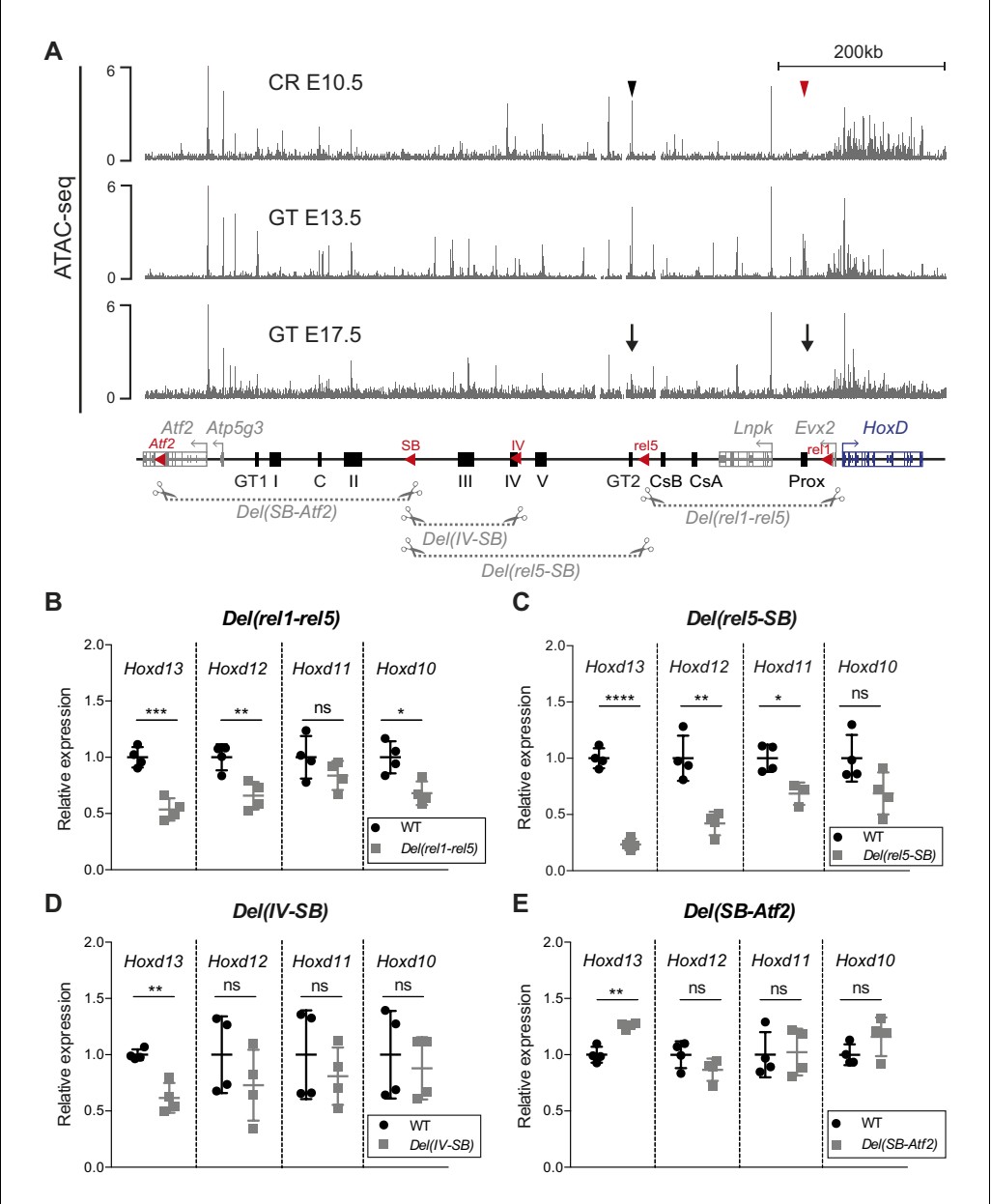

**Figure 3.** Various segments of C-DOM contribute to *Hoxd13* transcription in the GT. (**A**) The gray tracks show ATAC-seq profiles of E10.5 CR (average of two biological replicates, track 1), E13.5 GT (average of three biological replicates, track 2) and E17.5 GT (average of two biological replicates, track 3). In track one the red arrowhead shows the Prox enhancer and the black arrowhead the GT2 enhancer. In track 3, arrows indicate the loss of chromatin accessibility at the Prox and GT2 enhancer regions. Coordinates (mm10): chr2: 73815520–74792376. A schematic representation of the *HoxD* cluster and the C-DOM is shown below with known enhancers as black boxes. The red arrowheads represent the deletions breakpoints. The four large deletion alleles analyzed are depicted as gray dashed lines with scissors. (**B–E**) RT-qPCR of posterior *Hoxd* genes mRNAs for wildtype and homozygous mutant deletion alleles using E12.5 GT. The mutant allele is indicated on top of each plot. The values plotted indicate the ratio of mRNA levels using wildtype as a reference (black dots) (n = 4 biologically independent wildtype or mutant GT). A Welch's *t*-test was used to evaluate the statistical significance of changes in gene expression. Bars indicate mean with SD, *p≤0.02; **p≤0.007; ***p≤0.0005, ****p≤0.0001; ns = non significant.

The online version of this article includes the following source data for figure 3:

**Source data 1.** RT-qPCR values of *hoxd* genes in wildtype and deletion alleles.

E17.5 GT, as development progressed, these peaks were lost in the C-DOM (*Figure 3A*, arrows) correlating with the decrease in both *Hoxd* transcript levels and chromatin interactions (see above).

We evaluated the functional importance of C-DOM for the transcriptional control of *Hoxd* genes during GT development by using a series of partial deletions, in particular the *Del(rel1-rel5)*, *Del (rel5-SB)* and the *Del(SB-Atf2)* alleles (*Montavon et al., 2011*; *Figure 3A*, bottom), as well as the *Del (IV-SB)* allele corresponding to a deletion between island IV and SB (*Figure 3A*, bottom). The latter allele, a 155 kb large deficiency, removed half of the regulatory region between the *rel5* and *SB* breakpoints and contained three GT regulatory regions, E1, IIIE and IVE (see below). We analyzed the effect of each of these four deletions on *Hoxd* genes transcription by RT-qPCR at E12.5.

In the *Del(rel1-rel5)* allele, one-third of C-DOM is removed, including two digit and/or GT enhancers (GCR and Prox) (*Gonzalez et al., 2007*; *Spitz et al., 2003*; *Figure 3A*, bottom). In these mutant mice, a 47% reduction in *Hoxd13* mRNA levels was scored in the GT (p=0.0005), whereas *Hoxd12*, *Hoxd11* and *Hoxd10* were less affected (*Figure 3B* and *Figure 3—source data 1*). The *Del (rel5-SB)* allele is a 300 kb large deletion of C-DOM including the GT2, and the island III, IV and V regulatory sequences. Mice carrying this deletion displayed a greater effect on the steady-state level *Hoxd13* mRNAs, which was reduced by 76% (p<0.0001). Again, *Hoxd12*, *Hoxd11* and *Hoxd10* were affected to a lower extent (*Figure 3C* and *Figure 3—source data 1*). We next analyzed the *Del(IV-SB)* allele and noticed a 38% decrease in the amount of *Hoxd13* mRNAs (p=0.0066), yet no significant effect was detected for any other genes (*Figure 3D* and *Figure 3—source data 1*). Finally, we looked at the *Del(SB-Atf2)* allele where the most centromeric part of the TAD had been deleted. In these mutant mice, we observed a slight but significant upregulation of *Hoxd13* mRNA levels (p=0.003) in the GT, whereas other genes were not affected (*Figure 3E* and *Figure 3—source data 1*). Taken together, these results indicated that several non-overlapping regions located within C-DOM are required for the transcriptional activation of *Hoxd13* in the developing GT.

## Deletion of the Prox enhancer sequence

Within the different DNA intervals delimited by this former set of deletions, we assessed the contribution of single regulatory elements to the control of *Hoxd13* transcription. We applied CRISPR/ Cas9 genome editing to fertilized eggs and generated a series of alleles where these elements were either deleted or inverted. We initially focused on the region between the *rel1* and *rel5* breakpoints (*Figure 3A*, bottom and *Figure 4A*). In this genomic interval the limb- and GT-specific Prox enhancer (*Figure 4B*) accounted for the majority of chromatin interactions with *Hoxd13* and was accessible in the GT at E13.5 as seen by ATAC-seq (*Figure 3A*). We generated the *Del(Prox)* allele, a 13 kb large deletion including the Prox sequence (*Figure 4A*), and observed a 36% decrease in the expression of *Hoxd13* by RT-qPCR in E12.5 GTs (p=0.006) (*Figure 4C* and *Figure 4—source data 1*). This severe impact seemed to be exclusively quantitative, as the *Hoxd13* expression pattern detected by whole mount in situ hybridization (WISH) remained spatially unchanged (*Figure 4D*). This result indicated that the Prox enhancer accounts for more than a third of the *Hoxd13* transcriptional efficiency and is thus a major contributor to this regulation in GT.

We then looked at whether this effect was 'enhancer-autonomous' or if it involved a significant reorganization of the entire C-DOM regulatory landscape by performing ATAC-seq and 4C-seq in both control and *Del(Prox)* mutant E13.5 GTs (*Figure 4E–F*). The ATAC-seq profiles revealed no obvious change in chromatin accessibility throughout the C-DOM after the deletion of Prox (*Figure 4E*). Likewise, when we examined the potential importance of Prox in building the C-DOM interaction landscape by 4C-seq using *Hoxd13* as a viewpoint, we noticed only minor alterations in the frequency of contacts between *Hoxd13* and discrete *cis*-regulatory elements (*Figure 4F*). We thus concluded that the Prox enhancer, while of critical importance for regulating *Hoxd13*, does not actively contribute to the general architectural organization of the locus.

## Identification of GT-specific enhancers

We next focused on the genomic interval positioned between the *SB* and the *rel5* breakpoints (*Figure 5A*), since this region accounted for 76% of *Hoxd13* expression in the incipient genital bud (see *Figure 3C*). Based on ATAC-seq, H3K27ac ChIP-seq, 4C-seq datasets and on DNA sequence conservation, we selected five sub-regions on fosmid clones of approximately 40 kb in size and tested them for enhancer activity in transgenic assays (*Figure 5B,C*). Each region was cloned

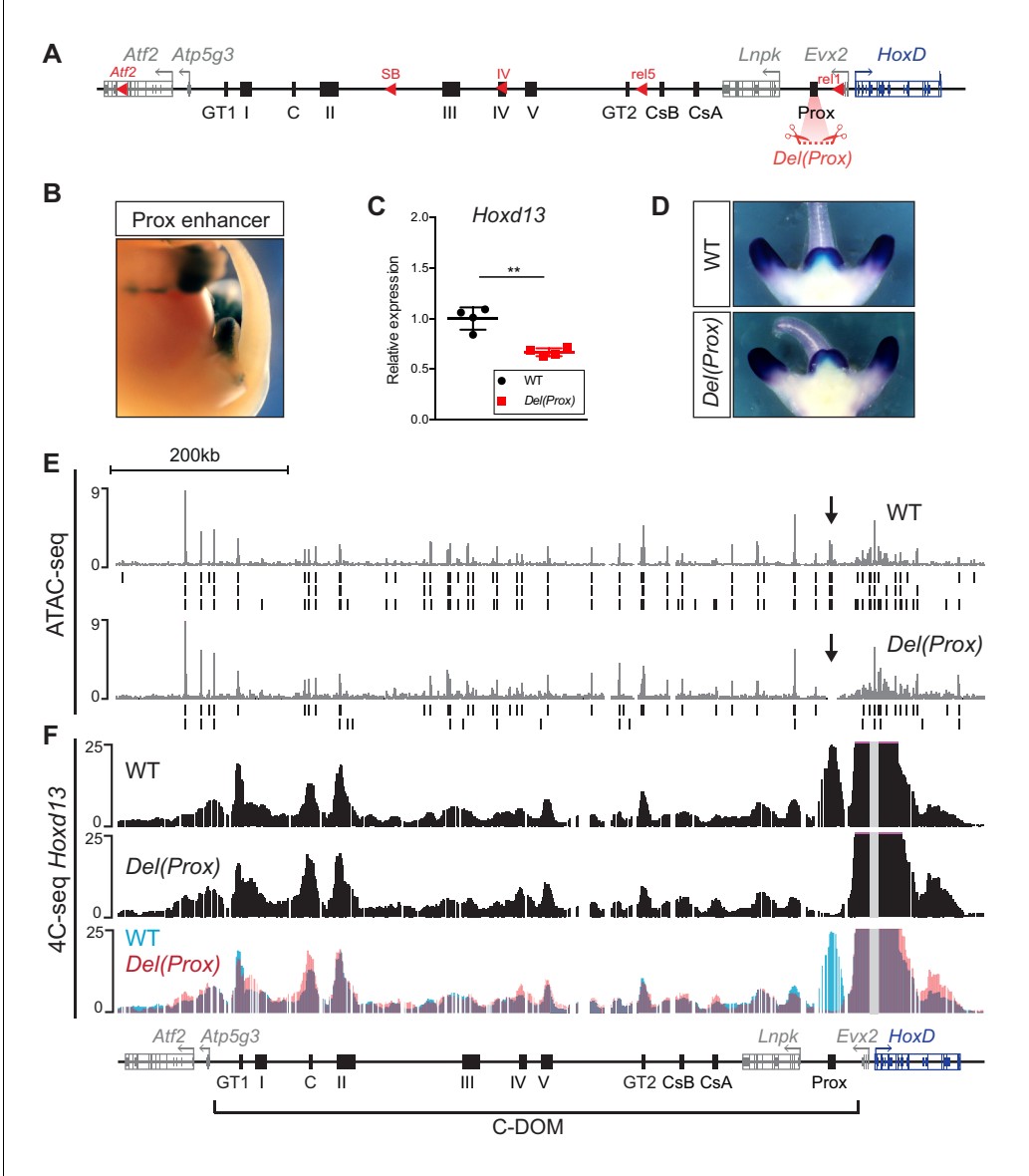

**Figure 4.** Deletion of the Prox enhancer. (**A**) Schematic representation of the *HoxD* cluster and the C-DOM with the deletion of the Prox sequence (*Del(Prox)*). (**B**) X-gal staining showing the activity of the Prox enhancer at E13.5. (**C**) *Hoxd13* transcripts levels obtained by RT-qPCR using wildtype and homozygous *Del(Prox)* mutant GTs at E12.5. The values plotted indicate the ratio of expression using wildtype as a reference (black dots) (n = 4 biologically independent WT or mutant GTs). A Welch's *t*-test was used to evaluate the statistical significance expression changes. Bars indicate mean with SD, **p=0.006. (**D**) WISH using the *Hoxd13* probe in both wildtype and *Del(Prox)* mutant E12.5 embryos. The *Hoxd13* expression pattern remained unchanged. (**E**) ATAC-seq profiles covering C-DOM and *HoxD* in wildtype (top) and *Del(Prox)* mutant (bottom) E13.5 GTs. Coordinates (mm10): chr2:73815520–74792376. The wildtype profile is the average of three biological replicates whereas the *Del(Prox)* represents the average of two biological replicates. Peaks called using MACS2 are displayed under the corresponding tracks (vertical black lines) for each individual replicate. Black arrows highlight the deleted region. (**F**) 4C-seq profiles (average of two biological replicates) of wildtype and *Del(Prox)* mutant E13.5 GTs. The Hoxd13 viewpoint is shown as a gray line. The overlay of the two wildtype (blue) and *Del(Prox)* (red) tracks highlight the loss of the Prox enhancer in the *Del(Prox)* allele and the lack of major alterations in the frequency of contacts between *Hoxd13* and discrete *cis*-regulatory elements. Coordinates (mm10): chr2:73815520–74792376.
The online version of this article includes the following source data for figure 4:

**Source data 1.** RT-qPCR values of *Hoxd13* in wildtype and *Del(Prox)* mutant GTs.

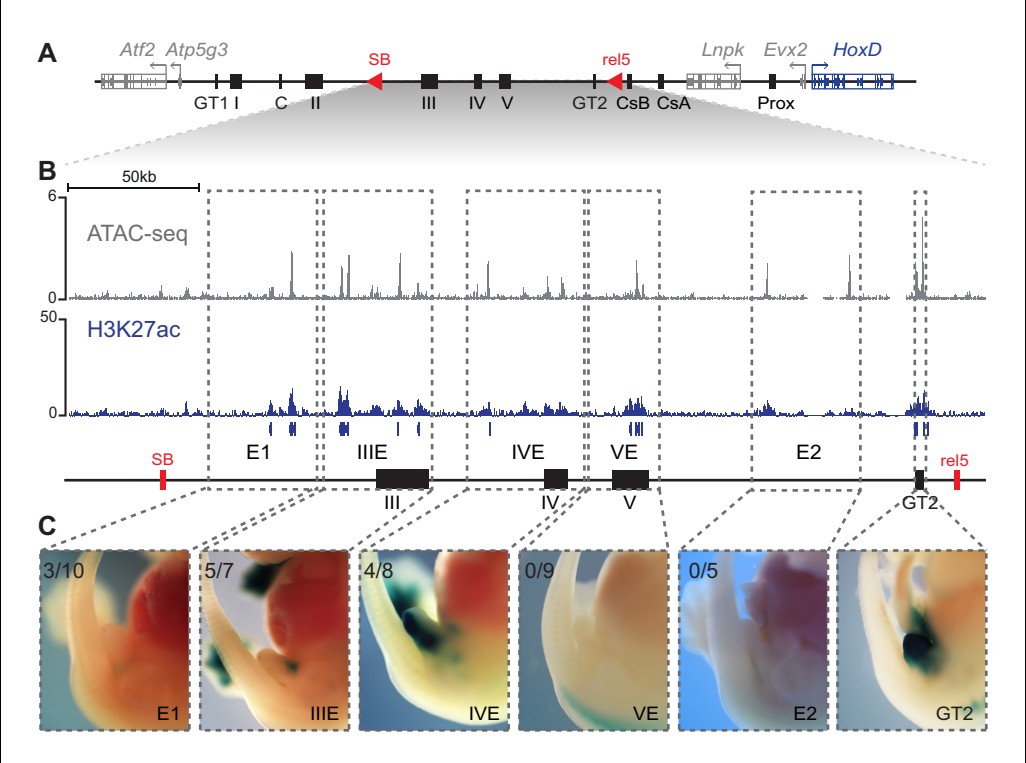

**Figure 5.** Activity of C-DOM regulatory elements in vivo. (**A**) Schematic representation of C-DOM and the *HoxD* cluster. Previously characterized enhancers are shown as black boxes and red arrowheads point to the *SB* and *rel5* breakpoints. (**B**) ATAC-seq profile (top, average of three biological replicates) and H3K27ac ChIP-seq profile (bottom) of E13.5 GTs, focusing on the DNA interval between *rel5* and *SB* (coordinates mm10: chr2:74084880–74432824). The vertical blue lines below the H3K27ac ChIP-seq profile represent the output of the MACS2 peak caller tool using the corresponding input as control. (**C**) Enhancer transgene activity of all the individual regulatory sub-regions analyzed within the *rel5* to *SB* interval. The gray dashed line boxes represent the tested sub-regions as well as the GT2 sequence. For each clone, a representative staining is shown at E13.5. The number of embryos showing lacZ reporter activity in the GT over the total number of embryos with an integrated transgene is indicated on the top left corner.

upstream of a *lacZ* reporter gene driven by a minimal *beta-globin* promoter and integrated at random positions in the mouse genome.

X-gal staining of E13.5 transgenic embryos revealed enhancer activity in the GT for the IIIE and IVE sequences (*Figure 5C*), in cellular territories included within the wider expression domain of *Hoxd13* in this structure. These two sequences showed complementary specificities, with IIIE active in dorsal GT cells, whereas the IVE sequence strongly labelled the ventral half of the GT (*Figure 5C*). Embryos injected with the E1 sequence showed a weak signal on the GT (*Figure 5C*) and no staining was scored either when using the VE, or the E2 sequences (*Figure 5C*), despite their promising chromatin signatures. Of particular interest, the VE region includes a CTCF binding site and this particular sequence is the only strongly occupied CTCF site present in the ca. 550 kb large region between *Evx2* and island II.

Therefore, out of the five regions tested, only E1, IIIE, and IVE showed some activity in the developing GT. We also re-investigated the activity of the GT2 sequence in transgenic embryos and scored a strong staining throughout the bud (*Figure 5C*). These experiments highlighted the regulatory complexity of the C-DOM, with individual enhancer elements displaying distinct and complementary patterns of activity (e.g., IIIE and IVE), while others show largely overlapping domains of expression (e.g., GT2).

## Serial deletions of single *cis*-regulatory elements

To further evaluate the regulatory potential of these DNA sequences, we generated deletion alleles for all presumptive enhancers located between the *rel5* and *SB* breakpoints for, when deleted, this region had the largest impact upon *Hoxd13* transcription (*Figure 3C*). Independent mouse strains were thus produced carrying either a *Del(GT2)* (3 kb large deletion), *Del(IV)* (11 kb) or *Del(IIIE)* (39 kb) allele. In addition, to assess the importance of bound CTCF proteins within island V, we both deleted and inverted this 14 kb large region (*Del(V)* and *Inv(V)*, respectively; *Figure 6A*). As a read out, we quantified *Hoxd13* mRNA levels by RT-qPCR and examined the transcript distribution by WISH. Unexpectedly, we did not detect any significant difference, either in transcript levels or in their spatial patterns, in any of the *Del(GT2)*, *Del(IV)*, *Del(IIIE)*, *Del(V)* and *Inv(V)* alleles (*Figure 6B,C* and *Figure 6—source data 1*). Unlike the Prox sequence analyzed above, these results suggest that none of these sequences is in itself functionally important enough to elicit a visible transcriptional effect upon the main target gene, at least in the GT and at the developmental stage considered.

The lack of any visible effect of the *Del(GT2)* allele was particularly surprising since this sequence displayed a strong, highly specific and continuous staining in the GT in transgenic embryos and also because of the robust transcriptional down-regulation obtained when using a larger deletion including it. Consequently, we performed both 4C-seq and ATAC-seq in *Del(GT2)* homozygous GT at E13.5 to assess whether this deletion would at least impact the functional organization of the regulatory landscape (*Figure 6—figure supplement 1*). Except for the loss of a single accessibility peak located between GT2 and CsB (*Figure 6—figure supplement 1A*, black arrows), the distribution of

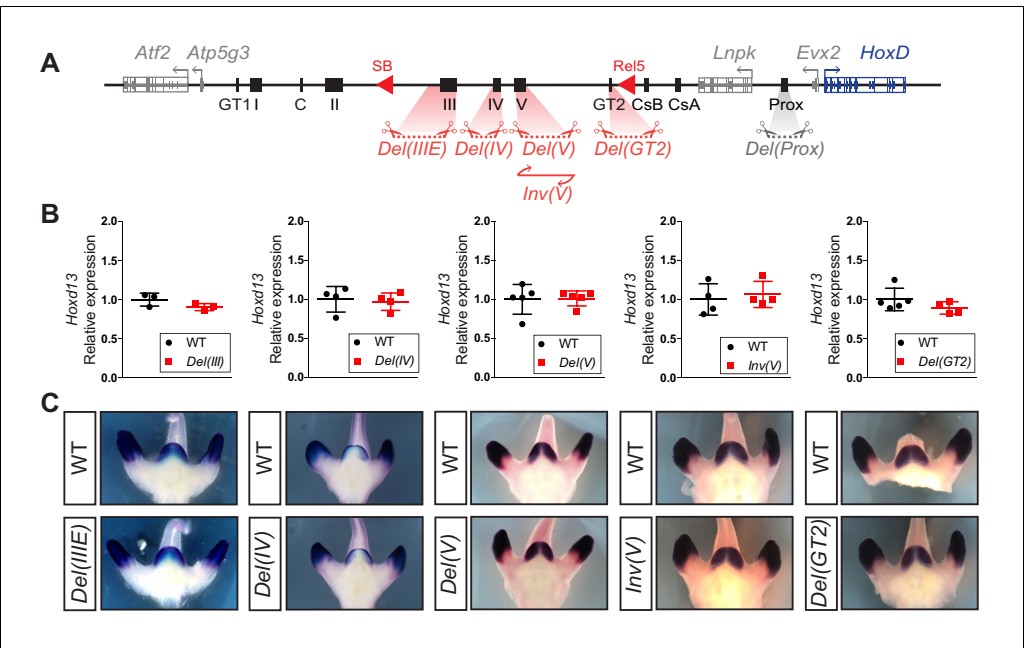

**Figure 6.** Serial deletions of single *cis*-regulatory elements. (**A**) Schematic representation of the alleles generated by CRISPR-Cas9 editing in vivo. The extents of sequences to be deleted were selected based on ATAC-seq, H3K27ac ChIP-seq, 4C-seq datasets and transgenic assays either obtained in this work or in *Lonfat et al. (2014)*; *Montavon et al. (2011)*. (**B**) Relative expression of *Hoxd13* obtained by RT-qPCR of both wildtype control and the various mutant alleles using E12.5 GT cells. The values plotted indicate the ratio of expression using wildtype as a reference (black dots) for each gene (n ≥ 3 biologically independent wildtype and mutant GTs). When using a Welch's *t*-test, no statistically significant change in expression was detected. (**C**) WISH using the *Hoxd13* probe and both wildtype and mutant E12.5 littermates. Both the mRNA levels and transcripts distribution remained globally unchanged.

The online version of this article includes the following source data and figure supplement(s) for figure 6:

**Source data 1.** RT-qPCR values of *Hoxd13* in wildtype and deletion alleles.
**Figure supplement 1.** Deletion of the GT2 enhancer.
**Figure supplement 2.** Alleles generated by CRISPR-Cas9.

accessible DNA sequences over C-DOM appeared to be independent from the GT2 element (*Figure 6—figure supplement 1A*). This absence of global impact of the GT2 deletion was confirmed when using a viewpoint on *Hoxd13* to evaluate by 4C-seq, potential reallocations of contacts in the mutant allele. There again, no salient change in the chromatin organization of C-DOM was observed (*Figure 6—figure supplement 1B*), further indicating that the deletion of GT2 in isolation had essentially no effect on the global architecture of the C-DOM landscape.

This absence of any detectable impact after deletion of a strong and specific enhancer as defined by all currently used parameters, can be due to a variety of reasons (see the discussion). Amongst them, the possibility that the functional contribution of GT2 is required at a particular stage of GT development, which was not considered in our analyses. To explore this possibility, we used RT-qPCR to measure the *Hoxd13* mRNA level in the CR at E10.5, a developmental stage where this enhancer is already accessible, as concluded from our ATAC-seq dataset (*Figure 6—figure supplement 1C*, black arrow), and capable of triggering *lacZ* transcription (*Lonfat, 2013*). At this early stage, we observed a slight (27%), but significant (p=0.0152) decrease in the expression of *Hoxd13* (*Figure 6—figure supplement 1D* and *Figure 6—source data 1*), suggesting that GT2 on its own may have a role in controlling *Hoxd13* expression prior to GT formation.

## CTCF and C-DOM chromatin organization

Amongst the various sequences isolated in C-DOM, island V was shown to interact with *Hoxd13* in all tissues and developmental stages analyzed thus far. We used our *Del(V)* and *Inv(V)* alleles to evaluate the importance of this element in ensuring proper 3D-chromatin organization at the *HoxD* locus. We first defined the CTCF binding profile in wildtype and mutant E13.5 GTs by using both ChIP-seq and CUT and RUN. In the wildtype locus, our ChIP-seq results showed several CTCF binding sites in the centromeric part of C-DOM, primarily between island II and *Atf2* and matching with other islands and 4C-seq peaks, close to the centromeric TAD boundary (*Figure 7A*). Of note, island V was the only region between *Evx2* and island II (approximately 550 kb) where a clear binding of this protein was detected (*Figure 7A*, arrow). A close examination of this element revealed a major CTCF binding site oriented towards the cluster and a weaker site observed nearby. In the *HoxD* cluster, the distribution of bound CTCF was similar to that observed in limb buds cells (*Rodríguez-Carballo et al., 2017*; *Soshnikova et al., 2010*), with a series of strong sites at its 5' extremity flanking *Hoxd13* and orientated towards C-DOM (*Figure 7A*).

We first verified the CTCF binding profiles in the two island V mutant alleles. As expected, the *Del(V)* allele showed a loss of the associated CTCF binding (*Figure 7A*). In contrast, strong CTCF binding to the major peak was detected in the *Inv(V)* allele in GT at E13.5, indicating that the inversion did not affect its binding capacity (*Figure 7A*). We looked at the potential impact of either deleting or inverting this CTCF site on the remaining regulatory elements by performing ATAC-seq in wildtype, *Del(V),* and *Inv(V)* homozygous GT at E13.5. In mutant *Del(V)* GT cells, with the exception of the deleted region, we did not observe any change in the ATAC-seq profile (*Figure 7—figure supplement 1A*) when compared to control GT cells. Minor changes were not reproduced in replicates and were likely due to individual variation (*Figure 7—figure supplement 1A*). In the mutant *Inv(V)* GT cells, we observed the loss of one ATAC-seq peak located between the GT2 and CsB sequences (*Figure 7—figure supplement 1A*, black arrow), similar to what was scored in the *Del(GT2)* allele. Therefore, neither the deletion nor the inversion of this centrally-located CTCF site had any substantial effect on the accessibility of the remaining regulatory elements, corroborating the RT-qPCR results where expression of *Hoxd13* was unchanged in these two alleles (*Figure 6*).

The position and orientation of this CTCF site suggested that it may play a role in helping the central part of C-DOM, rich in potential GT-specific elements, to get closer to *Hoxd13* through the formation of a large loop. We thus performed 4C-seq by using the *Del(V)* and *Inv(V)* mutant alleles on GT cells at E13.5 to investigate whether either the absence or the inversion of the CTCF site would affect the interaction landscape within C-DOM. When *Hoxd13* was taken as a viewpoint for the *Del(V)* allele, the global interaction profile between *Hoxd13* and C-DOM was virtually identical to control (*Figure 7B*). We confirmed this result by using a viewpoint positioned on island IV, located nearby island V. Only a slight reduction in the frequency of contacts between island IV and *Hoxd13* was scored (*Figure 7B*, black arrow). Therefore, island V and its associated CTCF site seem to have a marginal importance in maintaining the global chromatin structure of this regulatory landscape.

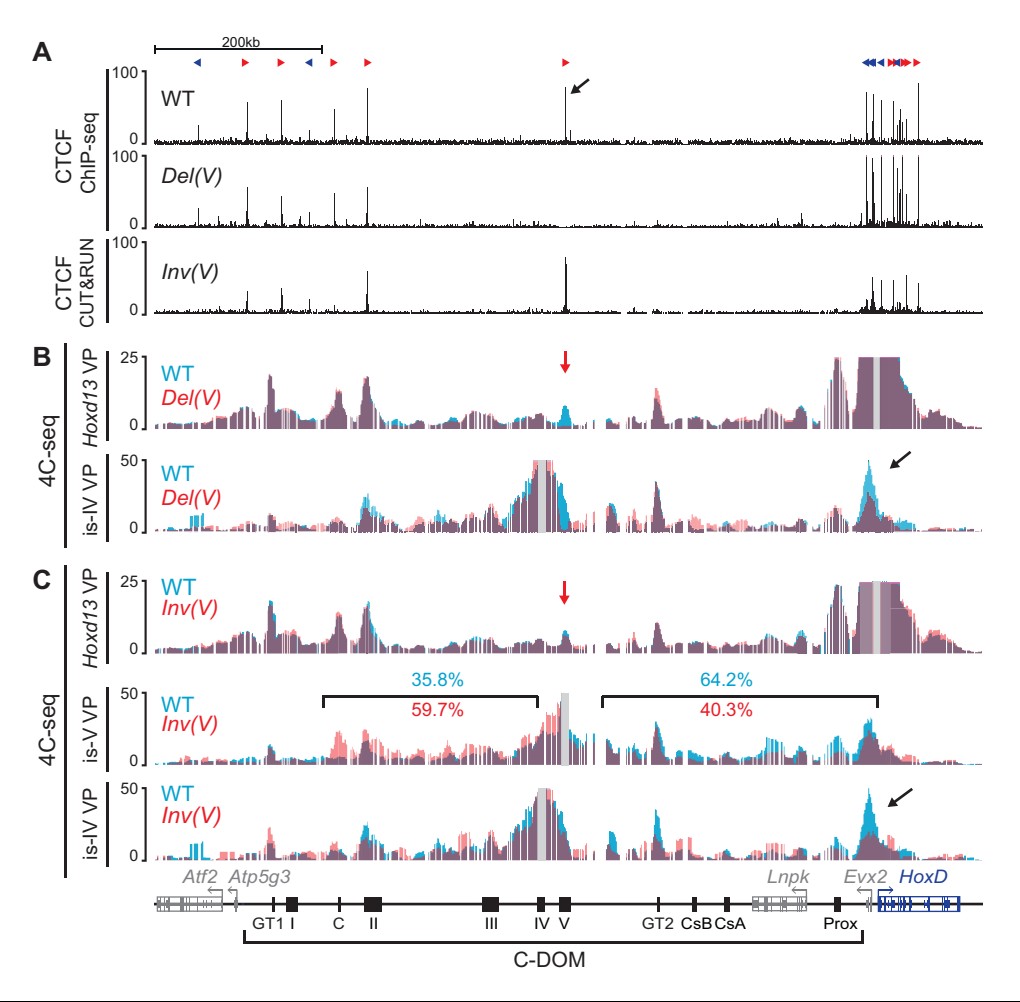

**Figure 7.** Deletion and inversion of the island V CTCF site in vivo. (**A**) CTCF ChIP-seq profiles of wildtype and *Del (V)* mutant E13.5 GTs. CUT and RUN of mutant *Inv(V)* E13.5 GT. The upper track shows the orientations of the CTCF motives (red and blue arrowheads). The black arrow indicates the major CTCF peak on island V. (**B**) 4C-seq profiles (average of two biological replicates) of wildtype and mutant *Del(V)* E13.5 GTs. The positions of the *Hoxd13* (upper tracks) and island IV (lower track) viewpoints are shown with a gray line. The profiles are displayed as overlays of wildtype (blue) and *Del(V)* (red). The red arrow shows the deleted region and the black arrow points to the *Hoxd13* region. (**C**) 4C-seq profiles (average of two biological replicates) of wildtype and mutant *Inv(V)* homozygous E13.5 GTs. Viewpoints are highlighted by a gray line. The profiles are shown as overlays of wildtype (blue) and *Inv(V)* (red). The red arrow shows the inverted region and the black arrow indicates the loss of contacts between island IV and the *Hoxd13* region in the *Inv(V)* sample. Percentages in blue (wildtype) and red (*InvV*) represent the proportion of the sums of interactions centromeric or telomeric to island V. (coordinates (mm10) for the quantifications: centromeric: chr2:74015789–74276083; telomeric chr2:74332870–74671433). Coordinates (mm10): chr2:73815520–74792376.

The online version of this article includes the following figure supplement(s) for figure 7:

**Figure supplement 1.** Chromatin accessibility in the deletion and inversion of the island V.

The majority of CTCF mediated chromatin loops are established between sites displaying opposite and convergent orientations (i.e. with CTCF motifs pointing toward each other) (*de Wit et al., 2015*; *Rao et al., 2014*; *Vietri Rudan et al., 2015*). Because our *Inv(V)* allele modified the orientation of this centrally-positioned CTCF site, we analyzed the impact of this inversion upon chromatin conformation. Qualitative analysis of the interaction profile generated using *Hoxd13* as a viewpoint revealed a slight disruption in the contacts between *Hoxd13* and island V (*Figure 7C*, red arrow). We validated this result by doing the reverse experiment and using a viewpoint on island V. In this set up, we observed a

reduction in the overall frequency of interactions in the region between island V and *Hoxd13* thus confirming the previous result (*Figure 7C*). Noteworthy, we observed an increase in the interaction frequency in the region centromeric to island V up to island II and island C, that is with the next CTCF sites displaying opposite and convergent orientations in the mutant configuration (*Figure 7C*, middle and *Figure 7—figure supplement 1B*). This inversion in the general tropism of interactions supports the importance of CTCF site orientations in the loop-extrusion model (*de Wit et al., 2015*; *Rao et al., 2014*; *Vietri Rudan et al., 2015*). When we used island IV as a viewpoint, we also observed a reduction in contacts with *Hoxd13* (*Figure 7C*, black arrow). Taken together, these results suggest that either the loss or the inversion of island V and its associated CTCF site, had an effect on C-DOM chromatin structure. Nonetheless, this effect did not greatly alter the regulatory landscape chromatin architecture, corroborating the lack of impact on transcription.

## Group 13 HOX proteins and TAD licensing

These results obtained after single enhancer deletions raise several hypotheses (see below). Among them is the possibility that the transcriptional outcome of the C-DOM regulation may rely upon an unspecific and global effect of accumulating various factors within the landscape architecture, thus licensing the TAD for activation of the target genes. The same C-DOM TAD was previously shown to regulate *Hoxd13* and neighboring *Hoxd* genes during distal limb bud development, a structure that resembles in many respects the developing genitals (*Cobb and Duboule, 2005*; *Cohn, 2011*; *Infante et al., 2015*; *Tschopp et al., 2014*). In this case, the products of both *Hoxa13* and *Hoxd13* were shown to bind to most of those C-DOM regulatory sequences specific for distal limb buds. From this observation, it was concluded that HOX13 proteins themselves were instrumental in activating or re-enforcing transcription of the *Hoxd13* gene in this developmental context, by accumulating at this landscape and binding to many accessible sites due to their low binding specificity (*Beccari et al., 2016*; *Sheth et al., 2016*).

In this context, we used an antibody against the HOXA13 in a CUT and RUN approach with either CR cells at E10.5 or GT cells at E13.5, that is before GT formation and during its emergence, respectively. Previous work has shown both redundancy of binding to limb regulatory elements and similarity of DNA binding motifs between HOXA13 and HOXD13 (*Sheth et al., 2016*). As such, and because of the HOXA13 binding profile in our dataset, we consider that this dataset reflects the binding of either HOXA13 or HOXD13, or of both proteins and is thus referred to as 'HOX13' (*Figure 8*). We detected enrichment of HOX13 binding signals in both *HoxD* regulatory landscapes (*Figure 8A*; C-DOM and T-DOM) similar to what was observed in distal forelimb (*Beccari et al., 2016*; *Sheth et al., 2016*). In CR cells at E10.5, HOX13 binding was found in C-DOM at discrete positions corresponding to previously described regulatory elements, in particular GT1, GT2, and Prox (*Figure 8B*). All these binding sites and others, with the exception of the Prox enhancer, correlated with accessible chromatin sites as mapped by ATAC-seq (*Figure 8B*, arrow). In the case of Prox, HOX13 binding was scored before a clear ATAC-seq signal was detected, suggesting a potential role for HOX13 proteins in participating to making some of these sites accessible. The few strong ATAC-seq peaks, which were not matched by HOX13 binding corresponded to non-*Hox* gene promoters (*Figure 8B*, bottom line).

In E13.5 GT cells, as development progressed in parallel with C-DOM becoming fully active, an overall increase of HOX13 binding was scored over C-DOM (*Figure 8B*). While binding was strengthened at some sites bound at the earlier stage, other elements became both accessible and bound by HOX13 such as the islands II and III regions or a sequence located inside an intron of the *Lnpk* gene (*Figure 8B*). Overall, a good correlation was observed between the increase of *Hoxd13* transcript levels on the one hand, and both the activation of the C-DOM regulatory landscape and the binding of HOX13, on the other.

## Discussion

### A preformed chromatin structure with multiple regulatory choices

In mammals, external genitals appear during fetal development as an overgrowth of a mesodermal territory surrounding the cloaca region (*Georgas et al., 2015*). In the absence of both *Hoxa13* and *Hoxd13* functions, this growth does not occur and the fetus displays a structure resembling that of a

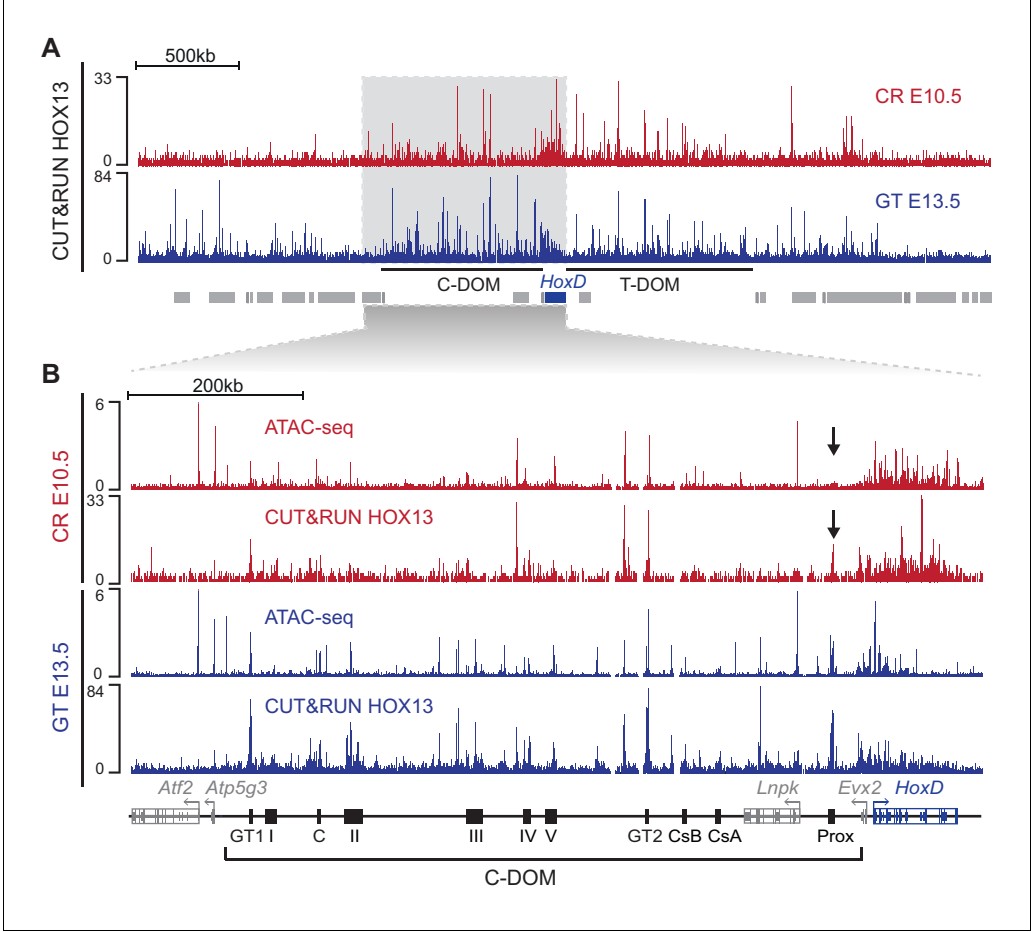

**Figure 8.** HOX13 protein binding in C-DOM. (**A**) HOX13 CUT and RUN profiles using E10.5 CR cells (red) and GT cells at E13.5 (Blue). The blue box represents the *HoxD* cluster and gray boxes are non-*Hox* genes. The profile encompasses 4 Mb and highlights the enrichment of HOX13 binding on both C-DOM and T-DOM *HoxD* regulatory landscapes. Coordinates (mm10): chr2: 72760109–76760109. (**B**) ATAC-seq and HOX13 CUT and RUN profiles of E10.5 CR cells (red) and E13.5 GT cells (Blue). Close-up view of C-DOM and the *HoxD* cluster (coordinates in mm10: chr2:73815520–74792376). The arrows indicate that although the Prox enhancer is bound by HOX13 in the CR at E10.5, the chromatin is not yet accessible at this element.

cloaca (*Kondo et al., 1997*; *Warot et al., 1997*), indicating that the proper transcriptional activation of these two genes in time and space is critical in this context. Studies of the *HoxD* cluster have provided some insights into this question (*Lonfat et al., 2014*) and suggested that the regulation of *Hoxd13* is primarily achieved by the C-DOM TAD, a large regulatory landscape flanking the gene cluster on its centromeric side, which also controls *Hoxd* gene activation in the developing digits. In the latter case, the chromatin interaction profile displayed some differences in transcriptionally active cells, even though the global TAD structure remained unchanged, suggesting that a C-DOM internal chromatin micro-organization had occurred due to the implementation of various digit-specific enhancers. Because of the related evolutionary origin of digits and external genitals (*Cobb and Duboule, 2005*; *Cohn, 2011*; *Tschopp et al., 2014*), we examined this particular aspect of *Hoxd* gene regulation during the growth of the genital tubercle.

We looked at chromatin dynamics at the *Hoxd* locus and observed two types of chromatin interactions. On the one hand, we detected contacts associated with a pre-formed structure, mainly linked to occupied CTCF sites. These contacts were observed independently of the transcriptional status of the cluster, as exemplified by island II and island V. On the other hand, we scored interactions present only when transcriptional activation had occurred such as the Prox and GT2 enhancer sequences. Our time-point series of interaction profiles revealed that the C-DOM TAD seems to be

activated in a coordinated manner, with all specific contacts appearing mostly within the same developmental time window, suggesting that the TAD itself may be considered as a global regulatory unit (see below), rather than a field containing a range of disparate enhancers with specific features and acting at different times. Also, the chromatin architecture associated with this specific developmental context was already observed in the E10.5 CR, that is before the emergence of the GT. Therefore, this internal-TAD micro-organization predates the outgrowth of the GT structure, which suggests -but does not demonstrate- a causal relationship or at least a necessity for the TAD to be fully primed for the structure to develop.

## Switching the TAD on and off to prevent regulatory leakages

Our time-series sampling gave us the unique opportunity to follow the C-DOM TAD dynamics in a developing system where most of the cells at E17.5 derive from a homogenous population of mesodermal cells in the nascent genital bud at E12.5, all expressing *Hoxd13*. The highest frequency of interactions with C-DOM was scored in E12.5 and E13.5 GTs, which correlated with an increase in *Hoxd13* transcription, an enrichment of H3K27ac marks and increase in binding of HOX13 proteins at discrete enhancer elements. After this time-point, a decrease in *Hoxd* transcript levels were scored in parallel with a reduction of all contacts associated with the active regulatory regions within the C-DOM. By E17.5 the C-DOM structure within the GT was reduced to a framework of constitutive interactions associated with CTCF binding sites, similar to the one observed in ES cells and fetal forebrain cells.

This global decrease, observed at a cell population level, can be explained either through a general decrease in transcription or through the selective transcriptional switch-off in some cell types along with their progressive differentiation thus leading to a dilution effect. Detailed ISH analyses see Figures 2 and 3 in *Dollé et al. (1991a)* and Figure 1 in *Warot et al. (1997)* clearly favors the latter option, whereby some cell types differentiating from the early mesodermal GT precursors turn off C-DOM regulation, whereas others maintain this regulation. At E17.5 indeed, strong *Hoxd13* expression was scored in the anlagen of the *corpus cavernosum* while other cells of the tubercle became negative. Of note, this concentration of positive cells in the blastema and subsequent restriction to the periphery (*Dollé et al., 1991a*) resembles the situation for *Hoxd13* transcripts during cartilage differentiation in developing digits. In support of this analogy, a penile bone (*baculum*) differentiates from this region in the mouse as in many other mammals.

The hereby described changes in the regulatory landscape architecture associated with transcriptional activity seem to be a pervasive feature during development (*Andrey et al., 2017*; *Freire-Pritchett et al., 2017*; *Phillips-Cremins et al., 2013*). We show that when these 'active' contacts disappear along with transcription being switched off, the TAD structure comes back to an inactive configuration. While such negative ground-state structures may simply reflect the absence of upstream factors and/or represent a scaffold to reinforce future enhancer-promoter contacts (*Paliou et al., 2019*), it may in our case be functionally required to prevent any transcriptional leakage of *Hoxd13*. HOX13 products are indeed potent dominant negative proteins (*Darbellay et al., 2019*; *Villavicencio-Lorini et al., 2010*) and their ectopic production in time and space must be prevented for proper development to be achieved (e.g. *Young et al., 2009*). A rapid return to an inactive chromatin conformation of C-DOM may help control this aspect, unlike other contexts where a particular chromatin topology is maintained for a long time (*Fernandez-Albert et al., 2019*).

## Long-range enhancers mechanism(s) of action

The complex pleiotropic expressions of vertebrate *Hox* genes, as well as of many other developmental genes, are usually controlled by multiple enhancers, either regulating subsets of the global pattern, or acting together in a partially redundant manner (*Long et al., 2016*; *Montavon et al., 2011*; *Spitz and Furlong, 2012*). We tested the potential function either of large DNA segments, or of shorter candidate regulatory regions within these segments and obtained different results depending on the position of the segment considered within C-DOM.

The deletion of the most centromeric part of C-DOM unexpectedly resulted in a slight but significant upregulation of *Hoxd13* expression. Various explanations can be considered for this result. Some sequences in this region may contact *Hoxd13* and thus prevent it to access more potent enhancers located at more proximal positions. This deleted region is rich in bound CTCF, which may favor such a sequestering (constitutive interactions) of *Hoxd13*. Upon deletion of this region,

*Hoxd13* contacts with functional enhancers would be re-enforced. Also, this effect may be a consequence of a global reorganization of the genomic landscape, for this specific deletion removes a TAD boundary. Because of the observed weak transcriptional effect and the difficulty to discriminate between many parameters, the analysis of this allele was not pursued further.

When the more central *rel5* to *SB* DNA fragment was deleted, a substantial decrease in *Hoxd13* transcription was observed. However, the deletion of any single candidate sequence in isolation identified within this segment did not elicit any detectable decrease in transcription. This systematic analysis echoes previous studies where deleting a single and well-characterized enhancer did not have the expected effect upon its target gene (e.g. *Cretekos et al., 2008*; *Frankel et al., 2010*; *Osterwalder et al., 2018*).

In contrast, the deletion of the Prox enhancer resulted in a decrease in *Hoxd13* transcripts, which in itself could account for the decrease observed when the larger *rel1* to *rel5* DNA fragment was deleted. This occurred in the absence of any major reorganization either of the chromatin architecture, or of its accessibility to factors. Therefore, Prox seemed to act independently of the other elements in C-DOM, as initially expected for a 'classical' enhancer sequence. Concerning the elements located within the *rel5* to *SB* central part of C-DOM whose deletions in isolation had no detectable effect, they could be functionally redundant with one another or, alternatively, compensatory mechanisms could be implemented for instance to re-direct the lost interactions towards another enhancer. Also, evolution might have selected regulatory processes to cope when facing particular conditions not necessarily tractable in laboratory conditions (*Frankel et al., 2010*; *Hong et al., 2008*). Our transgenic assays revealed that at least partial overlap in the functional domains was sometimes observed (GT1, GT2 and Prox), whereas in other cases, transgenic sequences elicited complementary domain of expression (IIIE and IVE). Therefore, some functional overlap between enhancers may account for the absence of phenotype (*Osterwalder et al., 2018*). Finally, it is possible either that our experimental approach lacks the resolution required to discern mild alterations in gene expression, perhaps occurring in a subpopulation of cells, or that individual C-DOM enhancers elements may control gene expression at distinct developmental stages. In the latter scenario, we may have missed the enhancer function by focusing our analyses in selected developmental time-point. Support for this alternative was provided by our results showing a decrease of *Hoxd13* mRNA in *del(GT2)* CR at E10.5.

## HOX13 proteins, super-condensates and pioneer effect

Besides these potential explanations, the binding of HOX13 proteins to most -if not all- of these C-DOM regulatory sequences raise yet another potential explanation related to recent work showing that phase-separation-induced condensates of RNA Pol II, transcription factors (TF) and the Mediator complex are present at particular enhancers leading to transcriptional activation (*Boija et al., 2018*; *Hnisz et al., 2017*; *Sabari et al., 2018*). In this view, condensate formation would be beneficial for transcriptional activation and could be promoted by the aggregation of protein containing intrinsically disordered regions (*Kato et al., 2012*). Both HOXD13 and HOXA13 contain long stretches of monotonic amino-acids (poly-Ala, Poly-Glu, Poly-Ser) (*Akarsu et al., 1996*; *Mortlock and Innis, 1997*; *Muragaki et al., 1996*), which could thus contribute to the building of this micro-environment by using the TAD as a scaffold, forming a sort of a super-condensate. Naturally-occurring modifications in the lengths of these amino-acids repeats were shown to drastically affect the function of HOX13 proteins (*Bruneau et al., 2001*; *Muragaki et al., 1996*; *Utsch et al., 2002*). Yet their effects upon a potential regulatory structure has not yet been evaluated. Binding of HOX13 proteins over C-DOM involved most -yet not all- sequences characterized as accessible by ATAC-seq. In the case of the Prox sequence a robust association was detected by CUT and RUN before an ATAC-seq peak could be scored, in support of the idea that HOX13 protein may in some instances display a pioneer effect (*Bulajić et al., 2019*; *Desanlis et al., 2019*).

## CTCF and the loop extrusion model *in embryo*

The *Rel5-SB* sub-region of C-DOM contains the largest series of defined GT regulatory sequences involved in *Hoxd13* regulation. Within this region lies island V, which contains the only occupied CTCF site in the central part of C-DOM. We initially assumed that this site could help increasing the spatial proximity between these enhancers and *Hoxd13* through looping due to the presence of several CTCF sites around *Hoxd13* with convergent orientations (*Rodríguez-Carballo et al., 2017*).

Also, this element is one of the two constitutive interactions maintained in the absence of transcription (along with island II). After inversion of island V and the CTCF site contained within, the effects upon the global chromatin architecture were marginal, consistent with the lack of transcriptional decrease observed upon deleting this element. All other identified regulatory sequences located nearby were still able to contact *Hoxd13* with the same profile, suggesting that this CTCF site had no major role in securing interactions between these enhancers and *Hoxd13*, similar to what was suggested at another developmentally regulated locus (*Williamson et al., 2019*).

The inversion of island V and its CTCF site nevertheless resulted in a global decrease of interactions with *Hoxd13*, balanced by an increase in interactions with the centromeric region containing distal CTCF sites. After inversion, these CTCF sites were now facing the island V CTCF binding site and hence these partial redistributions of interactions are in agreement with the loop extrusion model (*de Wit et al., 2015*; *Rao et al., 2014*; *Vietri Rudan et al., 2015*). While the inversion of island V thus resulted in a slight reallocation of intra-TAD interactions, they were not sufficient to elicit changes in gene expression and had negligible impact on long-range regulation of *Hoxd* genes by C-DOM. Alternatively, we may be missing the time and/or cellular resolution to observe the impact of removing these sites.

## Materials and methods

### Mouse strains and genotyping

Genotyping of all alleles was done by PCR. Mouse tissue biopsies were lysed for 15' at 95°C, 800 rpm, in lysis buffer (50 mM NaOH, 0.2 mM EDTA). For all genotyping reactions PCR was performed with a standardized cycling protocol (1x (94°3'), 2x (94°1', 62°1', 72°1'), 30x (94°30', 62°30', 72°30'), 1x (72°10')). The primers used to genotype the *Del(rel1-rel5)*, *Del(rel5-SB)* and *Del(SB-Atf2)* alleles can be found in *Montavon et al. (2011)*. Primers used to genotype the other alleles can be found in *Supplementary file 1*.

### CRISPR-Cas9

With the exception of the *Del(rel1-rel5)*, *Del(rel5-SB)* and *Del(SB-Atf2)* alleles (*Montavon et al., 2011*), all mouse strains carrying deletions or inversions of the different regulatory regions were generated using CRISPR–Cas9 genome editing technology. Single guide RNAs (sgRNAs) were designed 5' and 3' the genomic regions of interest using the crispr.mit.edu web tool (the Zang laboratory) for the *Del(V)*, *Inv(V)* and *Del(GT2)* alleles, or CCTop (*Stemmer et al., 2015*) for the *Del(IV)*, *Del(IIIE)*, *Del(Prox)* and *Del(IV-SB)* alleles (*Supplementary file 2*). All sgRNAs were cloned as recommended in *Cong et al. (2013)*, into the BbsI site of the pX330:hSpCas9 (Addgene ID 42230) vector. The mouse *Del(V)*, *Inv(V)* and *Del(GT2)* strains were produced by pronuclear injection (*Mashiko et al., 2013*) of a mix of the two appropriate sgRNAs cloned into the pX330:hSpCas9 vector (sgRNA:pX330:hSpCas9) (25 ng/µl each). The mouse *Del(IV-SB)*, *Del(IV)*, *Del(IIIE)* and *Del(Prox)* strains were produced by electroporation (*Hashimoto and Takemoto, 2015*) using a mix containing Cas9 mRNA (final concentration of 400 ng/µl) and two sgRNAs (300 ng/µl each) in Opti-MEM 1x injection buffer. PCR-based genotyping was carried out with primers designed on both sides of sgRNAs targets, with an approximate distance of 150 to 300 bp from the cutting site (*Supplementary file 1*). Sanger sequence of positive PCR bands was used to identify and confirm the deletion or inversion breakpoints of the F0 funder animals (*Figure 6—figure supplement 2*).

### Transgenic analysis

Murine fosmid clones were obtained from BACPAC Resources Center (https://bacpacresources.org) (*Supplementary file 3*). Their integrity was verified by Sanger sequence and restriction enzyme fingerprinting. The fosmids were introduced in EL250 cells (*Lee et al., 2001*) and targeted, by ET-recombineering with a construct containing a *PI-SceI* restriction site, a *β-globin::lacZ* reporter gene with a FRT-flanked kanamycin selection marker and flanked by 50 bp-long homology arms. The targeting constructs were produced by PCR amplification using the primers indicated in *Supplementary file 4* to introduce the homology arms. The WI1-1741G6 fosmid was shortened to remove the sequences that corresponded to island IV, to a final size of 28 kb (coordinates mm10 chr:74281022–74309212). The targeted fosmids were selected at 30°C on LB plates containing

chloramphenicol and kanamycin. The integrity of each modified fosmid was verified by restriction enzyme fingerprinting, and the correct integration of the *β-globin::lacZ* reporter gene was confirmed by PCR and Sanger sequence. All fosmids were linearized with *PI-SceI* and micro-injected into mouse oocytes. Embryos were harvested at E13.5 and stained for β-galactosidase activity following standard procedures. A minimum of three transgenic animals with consistent staining were obtained per construct. The transgenic mouse embryos for either the Prox or GT2 were obtained as described in *Gonzalez et al. (2007)*; *Lonfat et al. (2014)*. Embryos were stained using standard procedures. Briefly, whole embryos (E13.5) were fixed in 4% paraformaldehyde at 4°C for 35 min, stained in a solution containing 1 mg/ml X-gal at 37°C overnight, washed in PBS, and imaged.

## Whole-mount in situ hybridization

Whole-mount in situ hybridization (WISH) was performed according to *Woltering et al. (2014)*. Embryos were dissected in PBS and fixed overnight in 4% paraformaldehyde (PFA), washed in PBS, dehydrated, and stored in 100% methanol at −20°C. Rehydration was performed by a series of methanol/TBS-T washes, followed by a short digestion of Proteinase K, and re-fixation in 4% PFA. Pre-hybridization, hybridization, and post-hybridization steps were carried out at 67°C. For all genotypes, both mutant and control wildtype (E12.5) littermate embryos were processed in parallel to maintain identical conditions throughout the WISH procedure. DIG-labeled probes for in situ hybridizations were produced by in vitro transcription (Promega) and detection was carried out using an alkaline phosphatase conjugated anti-digoxigenin antibody (Roche). The *Hoxd13* WISH probe was previously described in *Dollé et al. (1991b)*. For detection, the chromogenic substrates NBT/BCIP or BM-purple were used.

## RT-qPCR

Before processing, all tissues were stored at −80°C in RNA*later* stabilization reagent (Invitrogen). RNA was extracted from single micro-dissected GT (E12.5) or single cloaca region (CR) (E10.5), using Qiagen Tissue Lyser and RNeasy Plus kit (Qiagen), according to the manufacturer's instructions. RNA was reverse transcribed using Superscript III (Invitrogen) or Superscript IV (Invitrogen) and random hexamers. qPCR was performed on a CFX96 real-time system (BioRad) using GoTaq qPCR Master Mix (Promega). Primers were previously described in *Montavon et al. (2008)*. Three technical replicates were used per biological replicate. Relative gene expression levels were calculated by the $2^{-\Delta Ct}$ method using a reference gene. *Tubβ* was chosen as internal control and the mean of wildtype control samples was set as reference to calculate the ratio between the different samples. Graphical representation and statistical analysis were performed with GraphPad Prism 7.

## 4C-seq

Circular chromosome conformation capture (4C-seq) was performed as described in *Noordermeer et al. (2011)*. Briefly, tissues (20 to 40 GT or 40 CR) were isolated in PBS supplemented with 10% Fetal Calf Serum and dissociated to single cell by collagenase treatment. Samples were fixed in 2% formaldehyde, lysed, and stored at −80°C. Samples were primarily digested with NlaIII (NEB, R0125L) followed by ligation under diluted conditions. After decrosslinking and DNA purification DpnII (NEB, R0543M) was used for the second restriction. All ligation steps were performed using highly concentrated T4 DNA ligase (Promega, M1794). For each viewpoint approximately 1 µg of DNA was amplified by using 12 individual PCR reactions. Libraries were constructed with inverse primers for different viewpoints (see *Supplementary file 5*) containing Illumina Solexa adapter sequences and sequenced on an Illumina HiSeq 2500 sequencer, as single-end reads (read length 100 base pairs or 80 base pairs). In some samples 4 bp barcodes were added between the adapter and each specific viewpoint to allow for sample multiplexing.

4C-seq reads were demultiplexed, mapped on GRCm38/mm10 mouse assembly, and analyzed using the 4C-seq pipeline of the Bioinformatics and Biostatistics Core Facility (BBCF) HTSstation (http://htsstation.epfl.ch) (*David et al., 2014*) or using a local version of it using the facilities of the Scientific IT and Application Support Center of EPFL. Profiles were normalized to a 5 Mb region surrounding the *HoxD* cluster and smoothened using a window size of 11 fragments. Quantifications of the interactions established with the cis-regulatory elements in *Figure 2C* were calculated as a percentage of the sum of the scores of each element using the mESC sample as a reference. Signals

falling either centromeric or telomeric to island V (in *Figure 7* and *Figure 7—figure supplement 1*) were assessed by calculating the sum of the scores in the region of interest normalized by the sum of the scores in both regions (coordinates mm10 for the quantifications: centromeric: chr2:74015789–74276083; telomeric chr2:74332870–74671433). All quantifications and bar plots were done using R (www.r-project.org).

## ChIP-seq

Micro-dissected GT (35 to 40) or CR (70) were crosslinked in 1% formaldehyde/PBS for 20 min and stored at −80°C until further processing. Chromatin was sheared using a water-bath sonicator (Covaris E220 evolution ultra-sonicator). Immunoprecipitation was done using the following antibodies, anti-CTCF (Active Motif, 61311), anti- H3K27ac (Abcam, ab4729), and anti-H3K27me3 (Merck Millipore, 07–449). Libraries were prepared using the TruSeq protocol, and sequenced on the Illumina HiSeq system (100 bp single-end reads) according to manufactures instructions.

ChIP-seq reads processing was done on the Duboule local Galaxy server (*Afgan et al., 2016*). Adapters and bad-quality bases were removed with Cutadapt version 1.16 (*Martin, 2011*) (options -m 15 -q 30 -a GATCGGAAGAGCACACGTCTGAACTCCAGTCAC). Reads were mapped to the mouse genome (mm10) using Bowtie2 (v2.3.4.1) (*Langmead and Salzberg, 2012*), with standard settings. The coverage was obtained as the output of MACS2 (v2.1.1.20160309) (*Zhang et al., 2008*). Peak calling in *Figure 5* was done using MACS2 (v2.1.0.20160309) callpeak (`—gsize` 1870000000) using the corresponding input data as control BAM (-c). CTCF motif orientation was assessed using the CTCFBSDB 2.0 database (*Ziebarth et al., 2013*), with EMBL_M1 identified motifs.

## ATAC–seq

ATAC–seq was performed as described in *Buenrostro et al. (2013)*. Micro-dissected tissues (a pool of 2 GT or 2 to 3 CR) were isolated in PBS supplemented with 10% Fetal Calf Serum and dissociated to single cell by collagenase treatment. After isolation, 50'000 cells were lysed in 50 μl of lysis buffer (10 mM Tris-HCl, pH 7.4, 10 mM NaCl, 3 mM MgCl$_2$ and 0.1% IGEPAL CA-630), nuclei were carefully resuspended in 50 μl transposition reaction mix (25 μl TD buffer, 2.5 μl Tn5 transposase and 22.5 μl nuclease-free water) and incubated at 37°C for 30 min. DNA was isolated with a MinElute DNA Purification Kit (Qiagen). Library amplification was performed by PCR (10 to 12 cycles) using NEBNext High-Fidelity 2x PCR Master Mix (NEB, M0541S). Library quality was checked on a fragment analyzer, and paired-end sequencing was performed on an Illumina NextSeq 500 instrument (read length 2 × 37 base pairs).

ATAC-seq reads processing was done on the Duboule local Galaxy server (*Afgan et al., 2016*). Reads were mapped to the mouse genome (mm10) using Bowtie2 (v2.3.4.1) (*Langmead and Salzberg, 2012*), (-I 0 -X 2000 `—fr` `—dovetail` `—very-sensitive-local`). Reads with mapping quality below 30, mapping to mitochondria, or not properly paired were removed from the analysis. PCR duplicates were filtered using Picard (v1.56.0). Peak calling was done using MACS2 (v2.1.0.20151222) callpeak (`—nomodel` `—shift` −100 `—extsize` 200 `—call-summits`). The coverage was done using the center of the Tn5 insertion and extended on both sides by 20 bp (script developed by L. Lopez-Delisle). When indicated, coverage profiles represent an average of the replicates, this was done by dividing each replicate by the number of million reads that fall within peaks in each sample (for normalization) and calculating the average coverage.

## RNA-seq

Micro-dissected GT from different embryonic stages were stored individually at −80°C in RNAlater stabilization reagent (Ambion) before further sample processing. Total RNA was extracted from tissues using Qiagen RNeasy Plus Micro Kit (Qiagen) after disruption and homogenization. RNA quality was assessed using an Agilent 2100 Bioanalyser. Sequencing libraries were prepared according to TruSeq Stranded mRNA Illumina protocol, with polyA selection. RNA-seq libraries were sequenced on an Illumina HiSeq 2500 sequencer, as single-end reads (read length 100 base pairs).

Raw RNA-seq reads were aligned on the mouse mm10 genome assembly using TopHat 2.0.9 (*Yates et al., 2016*). Gene expression computations were performed using uniquely mapping reads extracted from TopHat alignments and genomic annotations from filtered gtf from Ensembl release 82 (*Kim et al., 2013*) as described in *Amândio et al. (2016)*. FPKM (fragments per kilobase per

million mapped fragments) expression levels for each gene were calculated using Cufflinks (*Roberts et al., 2011*).

## CUT and RUN

CUT and RUN (*Schmid et al., 2004*; *Skene and Henikoff, 2017*) was performed as described in *Meers et al. (2019)*; *Skene et al. (2018)*. Micro-dissected tissues were isolated in PBS supplemented with 10% fetal calf serum and dissociated to single cell by collagenase treatment. After isolation, 500'000 cells were washed and bound to concanavalin A-coated magnetic beads and permeabilized with wash buffer (20 mM HEPES pH 7.5, 150 mM NaCl, 0.5 mM spermidine, and Roche Complete protein inhibitor) containing 0.02% digitonin. Bound cells were incubated with primary antibody (anti-HOXA13, AbCam ab106503; anti-CTCF, Active Motif, 6131) for 2 hr at room temperature. After washes the samples were incubated with Protein A-MNase (pA-MN) for 1 hr at 4° C, then washed twice more with Wash Buffer. Samples were resuspended in low-salt rinse buffer (20 mM HEPES, pH7.5, 0.5 mM spermidine, and 0.125% Digitonin) and chilled to 0°C and the liquid was removed on a magnet stand. Ice-cold calcium incubation buffer (3.5 mM HEPES pH 7.5, 10 mM CaCl$_2$, 0.05% Digitonin) was added and samples were incubated on an ice-cold block for 30 min. STOP buffer (270 mM NaCl, 20 mM EDTA, 4 mM EGTA, 0.02% Digitonin, 50 µg glycogen, 50 µg RNase A) was added and samples were incubated at 37°C for 30 min, replaced on a magnet stand and the liquid was removed to a fresh tube. DNA was extracted by Phenol-Chloroform extraction and ethanol precipitation. Libraries were prepared as described in *Skene et al. (2018)*. Library quality was checked on a fragment analyzer, and paired-end sequencing was performed on an Illumina NextSeq 500 instrument (read length 2 × 37 base pairs).

CUT and RUN reads processing was done on the Duboule local Galaxy server (*Afgan et al., 2016*). Adapters and bad-quality bases were removed with Cutadapt version 1.16 (*Martin, 2011*). Reads were mapped to the mouse genome (mm10) using Bowtie2 (v2.3.4.1) (*Langmead and Salzberg, 2012*), (-I 0 -X 1000 –fr –dovetail –very-sensitive). Reads with mapping quality below 30 or not properly paired were removed from the analysis. The output BAM file was converted to BED using bamtobed bedtools v2.18.2 (*Quinlan, 2014*). The coverage was obtained as the output of MACS2 (v2.1.1.20160309) (*Zhang et al., 2008*) (–format BED –keep-dup 1 –bdg –nomodel –extsize 200 –shift −100).

A complete list of datasets and replicate status of all experiments is described in *Supplementary file 6*.

## Ethics approval

All experiments were performed in agreement with the Swiss law on animal protection (LPA), under license No GE 81/14 (to DD).

## Acknowledgements

We thank Pierre Fabre for cloning one of the sgRNA used to generate the *Del(IV)* allele, Marion Leleu for her help with data analysis as well as members of the Duboule laboratories for insightful comments and discussion. We are grateful to Sandra Gitto and Thi Hanh Nguyen Huynh for their help with mice breeding and genotyping, as well as to Mylène Docquier, Brice Petit and Christelle Barraclough (University of Geneva), Bastien Mangeat, Elisa Cora and the GECF gene expression core facility (EPFL Lausanne) and the Lausanne genomic technologies facility (UNIL) for DNA sequencing.

## Additional information

### Funding

| Funder | Grant reference number | Author |
|---|---|---|
| Swiss National Science Foundation | 310030B_138662 | Denis Duboule |
| European Research Council | 588029 | Denis Duboule |

| National Institutes of Health | NICHD F32HD0935 | Christopher Chase Bolt |

The funders had no role in study design, data collection and interpretation, or the decision to submit the work for publication.

## Author contributions
Ana Rita Amândio, Conceptualization, Data curation, Formal analysis, Investigation, Methodology; Lucille Lopez-Delisle, Resources, Data curation, Software, Formal analysis; Christopher Chase Bolt, Bénédicte Mascrez, Investigation, Methodology; Denis Duboule, Conceptualization, Resources, Funding acquisition, Project administration

## Author ORCIDs
Ana Rita Amândio  https://orcid.org/0000-0003-0634-4042
Christopher Chase Bolt  http://orcid.org/0000-0002-3544-3552
Denis Duboule  https://orcid.org/0000-0001-9961-2960

## Ethics
Animal experimentation: All experiments were performed in agreement with the Swiss law on animal protection (LPA), under license No GE 81/14 (to DD).

## Decision letter and Author response
Decision letter https://doi.org/10.7554/eLife.52962.sa1
Author response https://doi.org/10.7554/eLife.52962.sa2

# Additional files

## Supplementary files
- Supplementary file 1. List of genotyping primers.
- Supplementary file 2. List of sgRNAs.
- Supplementary file 3. List of fosmids.
- Supplementary file 4. List of primers used for recombineering.
- Supplementary file 5. List of 4C-seq primers.
- Supplementary file 6. List of datasets and replicate status.
- Transparent reporting form

## Data availability
All raw and processed RNA-seq, 4C-seq, ChIP-seq, Cut & Run, and ATAC-seq datasets are available in the Gene Expression Omnibus (GEO) repository under accession number GSE138514.

The following dataset was generated:

| Author(s) | Year | Dataset title | Dataset URL | Database and Identifier |
|---|---|---|---|---|
| Amândio AR, Lopez-Delisle L, Bolt CC, Mascrez B, Duboule D | 2020 | A complex regulatory landscape involved in the development of external genitals | https://www.ncbi.nlm.nih.gov/geo/query/acc.cgi?acc=GSE138514 | NCBI Gene Expression Omnibus, GSE138514 |

The following previously published datasets were used:

| Author(s) | Year | Dataset title | Dataset URL | Database and Identifier |
|---|---|---|---|---|
| Noordermeer D, Leleu M, Schorderet P, Joye E, Chabaud F, Duboule D | 2014 | Temporal dynamics and developmental memory of 3D chromatin architecture at Hox gene loci | https://www.ncbi.nlm.nih.gov/geo/query/acc.cgi?acc=GSE55344 | NCBI Gene Expression Omnibus, GSE55344 |

| Amândio AR, Necsulea A, Joye E, Mascrez B, Duboule D | 2016 | Transcriptomic analysis of wild type and Del(Hotair)-/- mouse tissues | https://www.ncbi.nlm.nih.gov/geo/query/acc.cgi?acc=GSE79028 | NCBI Gene Expression Omnibus, GSE79028 |

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
