## [Decision Letter]

Thank you for submitting your article "A complex regulatory landscape involved in the development of external genitals" for consideration by *eLife*. Your article has been reviewed by three peer reviewers, including Job Dekker as the Reviewing Editor and Reviewer #1, and the evaluation has been overseen by Kevin Struhl as the Senior Editor. The following individual involved in review of your submission has agreed to reveal their identity: José Luis Gómez-Skármeta (Reviewer #3).

The reviewers have discussed the reviews with one another and the Reviewing Editor has drafted this decision to help you prepare a revised submission.

In this manuscript, Duboule and colleagues examine not only how complex regulatory architectures are established to achieve a specific transcriptional outcome, but also how they are rewired after. Using the *HoxD* locus and external genitalia as paradigm, the authors perform timecourse experiments to understand the correlation between structure and function and, importantly, generate a series of transgenic mouse lines to functionally disentangle the role of individual regulatory elements. Based on these observation they make the following conclusions: 1) the TAD regulatory environment of the *HoxD* locus is mostly unchanged during development, however specific regulatory elements become accessible and differentially interact with *Hoxd13* promoter upon gene activation; 2) while large deletions have strong effect on transcription, single enhancer deletions have surprisingly little effect on gene expression (with one notable exception) and 3) Inversion or deletion of CTCF contacts alters chromatin architecture landscape in agreement with predictions of the loop-extrusion model but has no effect on gene expression or enhancer-promoter contacts.

In general, the work is well done; the findings are interesting and novel and provide evidence that enhancer-promoter contacts are dynamically regulated within the general architecture of the encompassing TAD. The extensive transgenic mouse lines provide evidence that functional outcome of genetic manipulations cannot be readily predicted based on chromatin accessibility and 3D proximity and further confirmation that enhancer regulation can also be CTCF-independent. Overall, I believe that the manuscript could be potentially suitable for publication in *eLife* provided that the following important issues are addressed:

Essential revisions:

1) Replicates and biological reproducibility

In many figures the 4C signal is quantified (for example in Figure 2 and specifically in Figure 2C) and conclusions are made based on differences across conditions without showing what is the biological variability of such quantifications. Based on the Transparent Reporting File 2 biological replicates exist for these experiment – they should be analyzed separately to determine whether the differences are meaningful or due to biological variability.

2) Data presentation consistency

In many cases only selected timepoints are shown in the figure panels which sometimes makes the interpretation difficult. For example, in Figure 3A ATAC tracks from CR E10.5 and GT E13.5 are compared with H3K27ac tracks from GT E13.5 and GT E17.5. This makes it difficult to see if the accessibility at the Prox enhancer persists also after *Hoxd13* has been downregulated (at E17.5). The authors should display consistently the ATAC and H3K27ac tracks across all three timepoints for a fair comparison. Similar issue is in Figure 2B when we compare the 4C profiles between mES and CRE10.5 with gene expression in Figure 1B, which doesn't have the mES timepoint. Does expression of *Hoxd13* increase between mES and CR E10.5?

3) Contribution of the GT1 region to *Hoxd13* expression

The region labeled as GT1 in Figure 2 appears to interact strongly with *Hoxd13* promoter in a dynamic manner, is open at E13.5 but has no H3K27ac. Furthermore, in the *Del(SB-Atf2)* line, where the GT1 region is also removed, *Hox13* expression is upregulated. One hypothesis would be that this region represents a transcription repressor (potentially *Hox13* based on the ChIPseq in Figure 7), but the authors never discuss this. Such hypothesis could also potentially explain why the long deletion in the *Del(rel5-SB)* line has a major effect but there is little contribution of individual enhancer elements – it could be that the big deletion reduces the genomic distance between the *Hoxd13* promoter and GT1, therefore facilitating repression. It would be of great interest to see: 1) Does the interaction between *Hoxd13* and the GT1 region increase in the *Del(rel5-SB)* line? 2) Are there any motifs present in GT1, suggesting binding of repressors, is it marked by Polycomb? and 3) would the precise deletion/mutation of GT1 also lead to upregulation of *Hoxd13*.

4) In Figure 4: Is the expression of neighboring genes such as *Lnpk, Evx2* and *Atf2* (in the C-Dom) affected upon Prox deletion? This will address if Prox is regulating only *HoxD13* or has a general enhancer activity within the TAD. It would also help discriminated between additional layer of E-P specificity versus a global activation within the TAD. What is the expression of *Hoxd13* in this line at CR 10.5 and E17.5? Analogous for Figure 6 – does the deletion of GT2 affects the expression of neighboring genes? *Lnpk* is of particular interest, given the potential interaction based on Figure 2—figure supplement 1. Related to this point: Does inversion of the CTCF site in element lead to leakage of GT2 or Prox activity towards genes on the centromeric side of *HoxD*? It may not be surprising that deletion of element sC, II, III, IV by themselves do nothing, as they are "behind" the CTCF site in element V (they may in fact not regulate *Hoxd13* at all?). When element V is deleted: does deletion of these centromeric distal elements now alter *Hoxd13*?

5) A major conclusion is that the deletions do not affect the overall architecture of the locus. The question is whether 4C-seq, with just one or two anchors has enough power to conclude that there is no effect on the global architecture of the C-DOM landscape.

6) When GT2 or Prox are deleted: does *Hoxd13* now interact with other elements (new elements or elements it also interact with in WT but now with higher frequency)? If so, are these elements contained within the larger deletion that did have an effect on *Hoxd13* expression?

[Editors' note: further revisions were suggested prior to acceptance, as described below.]

Thank you for submitting your article "A complex regulatory landscape involved in the development of mammalian external genitals" for consideration by *eLife*. Your article has been reviewed by three peer reviewers, including Job Dekker at the Reviewing Editor and Reviewer #1, and the evaluation has been overseen by Kevin Struhl as the Senior Editor. The following individual involved in review of your submission has agreed to reveal their identity: José Luis Gómez-Skármeta (Reviewer #3).

The reviewers have discussed the reviews with one another and the Reviewing Editor has drafted this decision to help you prepare a revised submission.

Essential revisions:

The current 4C data represent only 1 replicate. Two reviewers indicated that it was critical to demonstrate reproducibility. They also understood the technical challenges. Therefore, we ask for adding biological replicates to confirm the most important and key findings. This would include replicates of some of the 4C analyses. When such data can be included the work would be suitable for *eLife*.

---

## [Author Response]

Essential revisions:1) Replicates and biological reproducibilityIn many figures the 4C signal is quantified (for example in Figure 2 and specifically in Figure 2C) and conclusions are made based on differences across conditions without showing what is the biological variability of such quantifications. Based on the Transparent Reporting File 2 biological replicates exist for these experiment – they should be analyzed separately to determine whether the differences are meaningful or due to biological variability.

We apologize for not being very clear regarding the number of replicates. We did produce replicates for all the deletion alleles and the corresponding controls (GT E13.5). The number of biological replicates for each condition (when more than one, see below) are indicated in the corresponding figure legends. However, we did not produce replicates for the samples used in the “developmental time series” (CRE10.5, GTE12.5, GTE15.5, GTE17.5). We determined it was not necessary in this specific case, mainly for two reasons. First because we made the converse experiment (i.e. using viewpoints on the islands) to confirm the changes. We understand that this is not the same as a biological replicate, yet the same result obtained by two (or more) different viewpoints is certainly as valid as running one duplicate for a single viewpoint. In the case where the material is rare and necessitates too many embryos (e.g. CRE10.5), we generally prefer using various viewpoints. Secondly, the time course of the GT in vivo (unique thus far in the literature) is self-controlled, i.e. the interactions that are not “specific” for one particular stage are used as internal comparative controls to identify those contacts that evolve along with the time course. This is particularly visible in Figure 2, where some peaks initially increase and subsequently decrease, when compared both vertically (time-scale) and horizontally (control interactions at the same time-point).

The Transparent Reporting File was corrected accordingly.

2) Data presentation consistencyIn many cases only selected timepoints are shown in the figure panels which sometimes makes the interpretation difficult. For example, in Figure 3A ATAC tracks from CR E10.5 and GT E13.5 are compared with H3K27ac tracks from GT E13.5 and GT E17.5. This makes it difficult to see if the accessibility at the Prox enhancer persists also after Hoxd13 has been downregulated (at E17.5). The authors should display consistently the ATAC and H3K27ac tracks across all three timepoints for a fair comparison. Similar issue is in Figure 2B when we compare the 4C profiles between mES and CRE10.5 with gene expression in Figure 1B, which doesn't have the mES timepoint. Does expression of Hoxd13 increase between mES and CR E10.5?

We tried to show the most significant datasets for the message we like to convey and thus we did not overpack the figures with confirmatory results. Nevertheless, we agree with this comment and we have re-organized the datasets to increase consistency. Concerning the first part of the question, we have now added to Figures 1 and 3 the ATAC-seq dataset for E17.5 for better comparison. We also removed the H3K27ac ChIP-seq data from Figure 3. Concerning the H3K27ac ChIP-seq for the cloaca, while we are confident in the results in the gene cluster (showed in Figure 1), we would need to do the experiment once again to determine that the C-DOM is virtually depleted of this mark. This experiment would require dissecting hundreds of young fetuses, which we do not think is necessary, nor desirable, considering the small amount of information that this experiment will deliver. As a consequence, we have removed the data from the figure.

To the second aspect of the question, the transcription of *Hoxd* genes increases from mES to the CRE10.5 sample, for there is essentially no expression of thesegenes in murine ES cells (5 FPKM for *Hoxd13* by RNA-seq, while undetected by qPCR according to the Noordermeer et al. paper (see Figure S7 from the *eLife,* 2014paper). Likewise, *Hoxd* genes are not transcribed in forebrain cells at any developmental stage considered and, consequently, there is little basis to quantifying *Hoxd13* either in mES cells, or in FB cells. On lines 188 and 189, we write: “These two profiles likely reflected the 3D chromatin state of C-DOM in the complete absence of transcription”, which we think is clear enough.

To clarify this issue, we added a schematic of the expression of *Hoxd13* in Figure 2. Also, we slightly amended Figure2A to make it clear that *Atf2* is not part of the C-TAD.

3) Contribution of the GT1 region to Hoxd13 expressionThe region labeled as GT1 in Figure 2 appears to interact strongly with Hoxd13 promoter in a dynamic manner, is open at E13.5 but has no H3K27ac. Furthermore, in the Del(SB-Atf2) line, where the GT1 region is also removed, Hox13 expression is upregulated. One hypothesis would be that this region represents a transcription repressor (potentially Hox13 based on the ChIPseq in Figure 7), but the authors never discuss this. Such hypothesis could also potentially explain why the long deletion in the Del(rel5-SB) line has a major effect but there is little contribution of individual enhancer elements – it could be that the big deletion reduces the genomic distance between the Hoxd13 promoter and GT1, therefore facilitating repression. It would be of great interest to see: 1) Does the interaction between Hoxd13 and the GT1 region increase in the Del(rel5-SB) line? 2) Are there any motifs present in GT1, suggesting binding of repressors, is it marked by Polycomb? and 3) would the precise deletion/mutation of GT1 also lead to upregulation of Hoxd13.

We certainly understand this interest for the GT1 sequence. We nevertheless decided to focus this work on the central and telomeric parts of C-DOM, rather than on the centromeric part containing GT1, mostly because the central deletion (and that of *Prox*) had the strongest impact on *Hoxd13* transcription. The situation turned out to be complex enough, with potential activating enhancers, such that we decided to ignore the up-regulation of *Hoxd* genes upon removal of the most centromeric deletion. The main reason for this is that the GT1 sequence was isolated because it displayed transgenic activity in the GT (LacZ staining). Therefore, should this sequence have a repressive activity, it would need to be a context-specific repression, whereas the sequence would be an activator outside its genomic context. While of potential interest, this investigation would require years of work using embryonic material and is in fact not on the agenda.

We nevertheless introduced a paragraph in the Discussion section “Mechanism(s) of action of long-range enhancers”, which now proposes some potential explanations related to the GT1-containing deletion.

Also, GT1 does not contain any salient DNA motive that would be known either to bind a repressor, or to induce formation of a chromatin structure that may be associated with transcriptional down-regulation (such as a TE). Along this line, the GT1 sequence does not display any histone modifications associated with polycomb binding.

4) In Figure 4: Is the expression of neighboring genes such as Lnpk, Evx2 and Atf2 (in the C-Dom) affected upon Prox deletion? This will address if Prox is regulating only HoxD13 or has a general enhancer activity within the TAD. It would also help discriminated between additional layer of E-P specificity versus a global activation within the TAD. What is the expression of Hoxd13 in this line at CR 10.5 and E17.5? Analogous for Figure 6 – does the deletion of GT2 affects the expression of neighboring genes? Lnpk is of particular interest, given the potential interaction based on Figure 2—figure supplement 1. Related to this point: Does inversion of the CTCF site in element lead to leakage of GT2 or Prox activity towards genes on the centromeric side of HoxD? It may not be surprising that deletion of element sC, II, III, IV by themselves do nothing, as they are "behind" the CTCF site in element V (they may in fact not regulate Hoxd13 at all?). When element V is deleted: does deletion of these centromeric distal elements now alter Hoxd13?

Several issues are raised here. First, one has to remember that the combined inactivation of *Hoxd13* and *Hoxa13* alone completely abrogates GT formation, leading to an open cloaca (see Introduction). To our knowledge, none of the genes located nearby (i.e. *Lnpk* and *Evx2*) has any function during either limb or GT development (the *Atf2* gene is not addressed here as it stands *outside* -and not *inside* as suggested- the C-DOM TAD). It is possible that these non-*Hox* genes nevertheless play a role by being part of the general enhancer-promoter equilibrium necessary for *Hoxd13* to be optimally regulated. The genetic stocks we use in this paper were not produced to address this question and only inferences can be obtained from our deleted mice. It would be necessary to produce mice lacking promoters (or the part responsible for the contact), a set of experiments largely outside the scope of this paper.

As it stands, we can propose the following answers:

In the *del(prox)* mice, the steady state mRNA levels of *Evx2* and *Hoxd12* are significantly decreased. This is not surprising as we showed previously (starting with Spitz et al., Cell, 2003) that every transcription unit present in this landscape would fall under the control of this set of enhancers (in fact, this is what defined the concept of regulatory landscapes in this paper). In this manuscript, we did not look at *Lnpk,* whose transcription is also controlled by the limb and genital enhancers present in C-DOM.

Concerning the expression of *Hoxd13* in the CR of *del(Prox)* embryos at E10.5, because this element is not accessible at this stage (as seen by ATAC-seq, Figures 3A and 8B), we considered that *Prox* was not active in these cells and hence we did not include this stage (which in addition, requires a large number of embryos as starting material) when analyzing this allele. Regarding the E17.5 stage, it would be difficult to interpret the results of this experiment (we expect only a moderate quantitative effect) as we will not be able to discriminate between a direct transcriptional effect, on the one hand, and a cellular dilution effect induced by the huge growth of the structure, with only some cells remaining positive for *Hoxd13* (see the text), on the other hand.

Also, the deletion of GT2 alone affects the transcription of *Evx2* but not that of *Hox* genes (there again, we did not look a *Lnpk*).

Does inversion of the CTCF site in element lead to leakage of GT2 or Prox activity towards genes on the centromeric side of HoxD?

There isn’t any detectable change in the expression of the *Evx2* and *Hoxd* genes in this CTCF inverted allele.

It may not be surprising that deletion of elements C, II, III, IV by themselves do nothing, as they are "behind" the CTCF site in element V (they may in fact not regulate Hoxd13 at all?).

Well, this does not seem to be the case, for we see an effect when deleting two elements (*III* and *IV)* at the same time while keeping the CTCF site associated to island V intact.

When element V is deleted: does deletion of these centromeric distal elements now alter Hoxd13?

This is of course one of the valid questions one may ask in experiments combining micro-deletions in-*cis*. We have not produced such alleles and thus we cannot provide a clear answer to this point. All this work was done in mice in vivo and regardless of how efficient is the CRISPR technology, such experiments require both an insane number of animals (to be well-done) and a long-time frame. Also, the number of possibilities of combined deletions in-*cis* is very large and there is no a priori one or a few particular combinations that should be done in priority, leaving us with the necessity either to have an unbiased approach or to have no approach at all. We selected the latter.

However, our educated guess is probably yes, because both the *del(IV-SB)* and the *del(rel5-SB)* alleles affect *Hoxd13* expression, although the first one keeps the island V intact and the second allele also removes GT2.

5) A major conclusion is that the deletions do not affect the overall architecture of the locus. The question is whether 4C-seq, with just one or two anchors has enough power to conclude that there is no effect on the global architecture of the C-DOM landscape.

We think that indeed our approach allows us to draw such a conclusion, provided the word “global” is used. It is certainly possible that minor changes occur that cannot be detected using our 4C approach, and the same question can be asked at any level of resolution. For example, a slight difference in the representativity of one particular spatial configuration amongst others would not even be detected by using a high resolution Hi-C approach. By global architecture we mean the presence of the TADs and of the main contact points within these TADs, suggesting that these various regulatory sequences do not contribute strongly to the building of the structure. Even the inversion of the CTCF site (*region V*), which induces a clear (and expected) modification in the distribution of interactions, does not globally change the contact profile to an extent similar form example to the inversion or the deletion of a strong TAD boundary region.

6) When GT2 or Prox are deleted: does Hoxd13 now interact with other elements (new elements or elements it also interact with in WT but now with higher frequency)? If so, are these elements contained within the larger deletion that did have an effect on Hoxd13 expression?

Unless we do not fully understand this comment, this point is addressed in the paper. For Prox in Figure 4E and F and in the text. For GT2, in Figure 6 and figure supplement 3A and 3B as well as in the text too.

[Editors' note: further revisions were suggested prior to acceptance, as described below.]

Essential revisions:The current 4C data represent only 1 replicate. Two reviewers indicated that it was critical to demonstrate reproducibility. They also understood the technical challenges. Therefore, we ask for adding biological replicates to confirm the most important and key findings. This would include replicates of some of the 4C analyses. When such data can be included the work would be suitable for eLife.

To the first point, there are two issues.

1) the appearance and disappearance, along with time, of particular peaks (GT2 and Prox; Figure 2). This is highly significant and we have tried to make it clearer in this version. For this time course, we had made one replicate at E13.5 to have an estimation of the variability. This variability is now indicated in the new Figure 2C (open circles in the E13.5 points). You can see that the data for GT2 are very clear. For Prox, you can see that if one applies this variability to all the points, only the time points 12.5, 13,5 and 15.5 are clearly significant, whereas the 10.5 and 17.5 points are borderline. This simply means that the Prox element peaks in its interactions within a shorter time window, which is in full agreement with the fact that it is opened (ATAC seq data) later than the GT2 sequence.

2) The general increase and decrease of contacts throughout the TAD (C-DOM), following the same dynamics. This was not the main issue of this Figure 2 and we agree that these exact percentages did not have much sense, except to indicate that there is a clear tendency. In this revised text, we removed the numbers from Figure 2 and simply mention in the text, after the description of GT2 and Prox dynamics that, in addition, the overall number of reads over the entire landscape showed a tendency… etc. A clear description of how we calculated this global increase/decrease in the number of reads has been added in the Materials and methods

To the second point, we have added a table including the 66 datasets described in this work. They are accompanied by the numbers of the figure(s) where they appear, the number of replicates (if any) and an explanation as to why replicates were not processed in some cases. This table is now included as supplementary file 6 and referred to at the end of the Materials and methods section as: “A complete list of datasets and replicate status of all experiments is described in Supplementary file 6”.